# Evolving MRSA: High-level β-lactam resistance in *Staphylococcus aureus* is associated with RNA Polymerase alterations and fine tuning of gene expression

**Viralkumar V. Panchal**[1,2], **Caitlin Griffiths**[3], **Hamed Mosaei**[3], **Bohdan Bilyk**[1,2], **Joshua A. F. Sutton**[1,2], **Oliver T. Carnell**[1,2], **David P. Hornby**[1], **Jeffrey Green**[1], **Jamie K. Hobbs**[2,4], **William L. Kelley**[5], **Nikolay Zenkin**[3], **Simon J. Foster**[1,2]*

1 Department of Molecular Biology and Biotechnology, University of Sheffield, Western Bank, Sheffield, United Kingdom, 2 The Florey Institute for Host-Pathogen Interactions, University of Sheffield, Sheffield, United Kingdom, 3 Centre for Bacterial Cell Biology, Biosciences Institute, Faculty of Medical Sciences, Newcastle University, Newcastle upon Tyne, United Kingdom, 4 Department of Physics and Astronomy, University of Sheffield, Sheffield, United Kingdom, 5 Department of Microbiology and Molecular Medicine, University Hospital and Medical School of Geneva, Geneva, Switzerland

* s.foster@sheffield.ac.uk

**Data Availability Statement:** All files containing raw data are available from the ORDA (DOI: https://doi.org/10.15131/shef.data.12383624). All other

## Abstract

Most clinical MRSA (methicillin-resistant *S. aureus*) isolates exhibit low-level β-lactam resistance (oxacillin MIC 2–4 μg/ml) due to the acquisition of a novel penicillin binding protein (PBP2A), encoded by *mecA*. However, strains can evolve high-level resistance (oxacillin MIC ≥256 μg/ml) by an unknown mechanism. Here we have developed a robust system to explore the basis of the evolution of high-level resistance by inserting *mecA* into the chromosome of the methicillin-sensitive *S. aureus* SH1000. Low-level *mecA*-dependent oxacillin resistance was associated with increased expression of anaerobic respiratory and fermentative genes. High-level resistant derivatives had acquired mutations in either *rpoB* (RNA polymerase subunit β) or *rpoC* (RNA polymerase subunit β') and these mutations were shown to be responsible for the observed resistance phenotype. Analysis of *rpoB* and *rpoC* mutants revealed decreased growth rates in the absence of antibiotic, and alterations to, transcription elongation. The *rpoB* and *rpoC* mutations resulted in decreased expression to parental levels, of anaerobic respiratory and fermentative genes and specific upregulation of 11 genes including *mecA*. There was however no direct correlation between resistance and the amount of PBP2A. A mutational analysis of the differentially expressed genes revealed that a member of the *S. aureus* Type VII secretion system is required for high level resistance. Interestingly, the genomes of two of the high level resistant evolved strains also contained missense mutations in this same locus. Finally, the set of genetically matched strains revealed that high level antibiotic resistance does not incur a significant fitness cost during pathogenesis. Our analysis demonstrates the complex interplay between antibiotic resistance mechanisms and core cell physiology, providing new insight into how such important resistance properties evolve.

relevant data are within the manuscript and its
Supporting Information files.

**Funding:** This work was supported by the 2022
Futures Initiative, University of Sheffield; Wellcome
Trust (212197/2/18/2, 102851/Z/13/Z and 217189/
Z/19/Z; https://wellcome.ac.uk/, SJF, JKH, NZ),
Medical Research Council (MR/T000740/1; https://
mrc.ukri.org/, NZ), Engineering and Physical
Sciences Research Council (EP/T002778/1; https://
epsrc.ukri.org/; SJF, JKH, NZ) and the Swiss
National Science Foundation (WLK 310030-
146540 and 10030-166611). The funders had no
role in study design, data collection and analysis,
decision to publish, or preparation of the
manuscript.

**Competing interests:** The authors have declared
that no competing interests exist.

## Author summary

Methicillin resistant *Staphylococcus aureus* (MRSA) places a great burden on human health-
care systems. Resistance is mediated by the acquisition of a non-native penicillin-binding pro-
tein 2A (PBP2A), encoded by *mecA*. MRSA strains are resistant to virtually all β-lactam
antibiotics, and can shift from being low- to high-level resistant. Prior studies have revealed
the involvement of components of the core genome in increased resistance, but the underlying
mechanism is still unknown. In this study, we have found that increased resistance is associ-
ated with mutations in either *rpoB* (RNA polymerase subunit β) or *rpoC* (RNA polymerase
subunit β') resulting in slower growth and elevated levels of PBP2A. Furthermore, transcript
profiling revealed that insertion of *mecA* triggered metabolic imbalance by altering anaerobic
and fermentative gene expression, accompanied by low-level resistance whereas, acquisition
of *rpoB* and *rpoC* mutations reversed gene expression to wild-type level and enabled cells to
become highly-resistant. The mutations also affected RNA polymerase activity. A set of
matched strains revealed that changes in antibiotic resistance levels do not have a significant
cost in terms of pathogenic potential. Our study reveals a novel effect of *mecA* acquisition on
central metabolism and sheds light on potential pathways essential for high-level resistance.

## Introduction

Infections caused by *Staphylococcus aureus* as well as methicillin-resistant *S. aureus* (MRSA)
pose a significant and enduring threat to human healthcare [1,2]. β-Lactams act by binding to
the transpeptidase domain of PBPs [3], whose activity is required for the final stages of bacte-
rial cell wall peptidoglycan biosynthesis [4]. In staphylococci, oxacillin (methicillin) resistance
is most commonly mediated by a penicillin-insensitive penicillin-binding protein (PBP2A)
encoded by *mecA* which has low affinity for β-lactams and maintains cell wall biosynthesis in
the presence of β-lactams, with the help of the native bifunctional penicillin-binding protein
(PBP) PBP2, for its transglycosylase activity [5].

Nucleotide sequence determination of the early MRSA isolate, N315 [6], identified a novel
mobile genetic element, staphylococcal cassette chromosome *mec* (SCC*mec*) which carries the
*mec* gene complex, regulatory components, surrounding ORFs and insertion sequences [7,8].
In all cases, SCC*mec* integrates into the *S. aureus* genome near the origin of replication at the
3' end of the *orfX* gene [9,10]. SCC*mec* also carries a cassette chromosome recombinase (*ccr*)
gene complex ensuring its mobility [11]. Currently, SCC*mec* elements are classified into types
I to XI on the basis of considerable structural variation in *ccr* and *mec* gene complexes [12].

To be resistant to methicillin, an MRSA strain must carry a functional copy of *mecA*, how-
ever, variations in temperature, osmolarity and pH alter the level of resistance conferred
[13,14]. Moreover, recent biochemical studies highlighted the importance of proper maturation
of PBP2A to exhibit broad β-lactam resistance [15]. Even though *mecA* is essential for resis-
tance, the level of resistance exhibited by clinical MRSA isolates varies greatly and this variation
cannot be simply due to the differences in the activity of regulatory genes/proteins (*mecR1/
mecI*) [16] as not all MRSA strains carry an intact *mec* regulatory system [17,18]. In addition,
isolates expressing low-level resistance (oxacillin MIC <2 μg/ml) exhibit heterogeneous expres-
sion of methicillin resistance, meaning that the majority of the bacterial population is sensitive
to low concentrations of antibiotic; however, a small proportion of cells exhibit high-level resis-
tance (>50 μg/ml) [14,16,19,20]. The frequency at which the highly resistant subpopulation
arises is about $10^{-4}$, which is reproducible and strain-specific [19]. Importantly, heterogeneously

resistant strains are capable of expressing high-level resistance upon exposure to elevated concentrations of β-lactam antibiotics or under specific growth conditions [13,14]. The phenotypic expression of high-level resistance is known as homogeneous resistance [14]. The transition from low- to high-level resistance has been shown to involve mutational events or genetic rearrangements to accommodate the resistance pathway. Conversion from low- to high-level resistance is complex, with mutations reported in genes associated with the stringent stress response (*relA*, *relQ*), *fem* (factors essential for methicillin resistance), auxiliary factors (*aux*), transcription (*rpoB*, *rpoC*), and c-di-AMP phosphodiesterase (*gdpP*); suggesting a network of processes are required for resistance [16,20–24]. However, the mechanism underpinning the conversion of low-level resistance into stable high-level resistance is ill-defined. Several studies to understand the molecular basis of methicillin resistance have been carried out, but all have either introduced multi-copy plasmid-borne *mecA* into a methicillin-sensitive isolate or used clinical MRSA isolates with different genetic backgrounds exposed to β-lactam antibiotics [16,20,22].

Here, we have used the well-characterised strain *S. aureus* SH1000 carrying a chromosomal copy of *mecA* to explore the mechanisms underpinning the acquisition of high-level methicillin resistance. This provides a new model system to examine the complexity of high-level resistance to β-lactam antibiotics.

## Results

### Construction and characterisation of highly oxacillin resistant derivatives of SH1000

Naturally occurring clinical MRSA isolates carry a single copy of *mecA* within the SCC*mec* cassette [7]. Therefore, we introduced *mecA* into the chromosome of SH1000 to better understand the mechanisms required to permit high-level resistance to methicillin. The pMUTIN4 [25] based insertion suicide plasmid pGM068 [26] was used to integrate *mecA* under its native promoter (p*mecA*) downstream of the *lysA* gene creating pVP01-p*mecA*. This plasmid contains a truncated 3´ region of a *lysA* gene encoding the terminal enzyme of the lysine biosynthesis pathway. Upon transformation of pVP01-p*mecA* into *S. aureus*, homologous recombination occurs creating an insertional duplication of the *lysA* region, conferring erythromycin resistance without lysine auxotrophy (Fig 1A).

The resultant "Untrained" strain *lysA*::p*mecA* (SJF4996) exhibited low-level oxacillin resistance (MIC 2 μg/ml) compared to the parental SH1000 (MIC 0.12 μg/ml) (Fig 1B). This level of resistance is similar to most clinical isolates of MRSA with heterogeneous resistance [27,28]. Highly oxacillin-resistant derivatives of *lysA*::p*mecA* (SJF4996) were subsequently isolated using gradient BHI media supplemented with 5 and 20 μg/ml methicillin (Fig 1C and 1D). This approach allowed selection of "Trained" mutants exhibiting oxacillin resistance levels from <2 to ≥256 μg/ml MIC (S1 Table). The *mecA* gene was then cured from a highly resistant derivative (SJF5003) by replacing *lysA*::p*mecA* with *lysA*::*kan* by phage transduction to give strain SJF5010 with an oxacillin MIC of 0.5 μg/ml. Subsequent reintroduction, by transduction, of *mecA* into *lysA*::*kan* (SJF5010) resulted in *lysA*::p*mecA*+ (SJF5011), with concomitant high-level oxacillin resistance (MIC ≥256 μg/ml). Thus, *mecA* alone cannot impart high-level resistance implying that chromosomal mutation(s) are also necessary.

### Conversion from low- to high-level oxacillin resistance is promoted by mutations in either *rpoB* or *rpoC*

Whole genome sequencing was performed on the parental strain (SH1000), *lysA*::p*mecA* (SJF4996) and 14 highly oxacillin-resistant derivatives (S2 Table). Apart from the desired

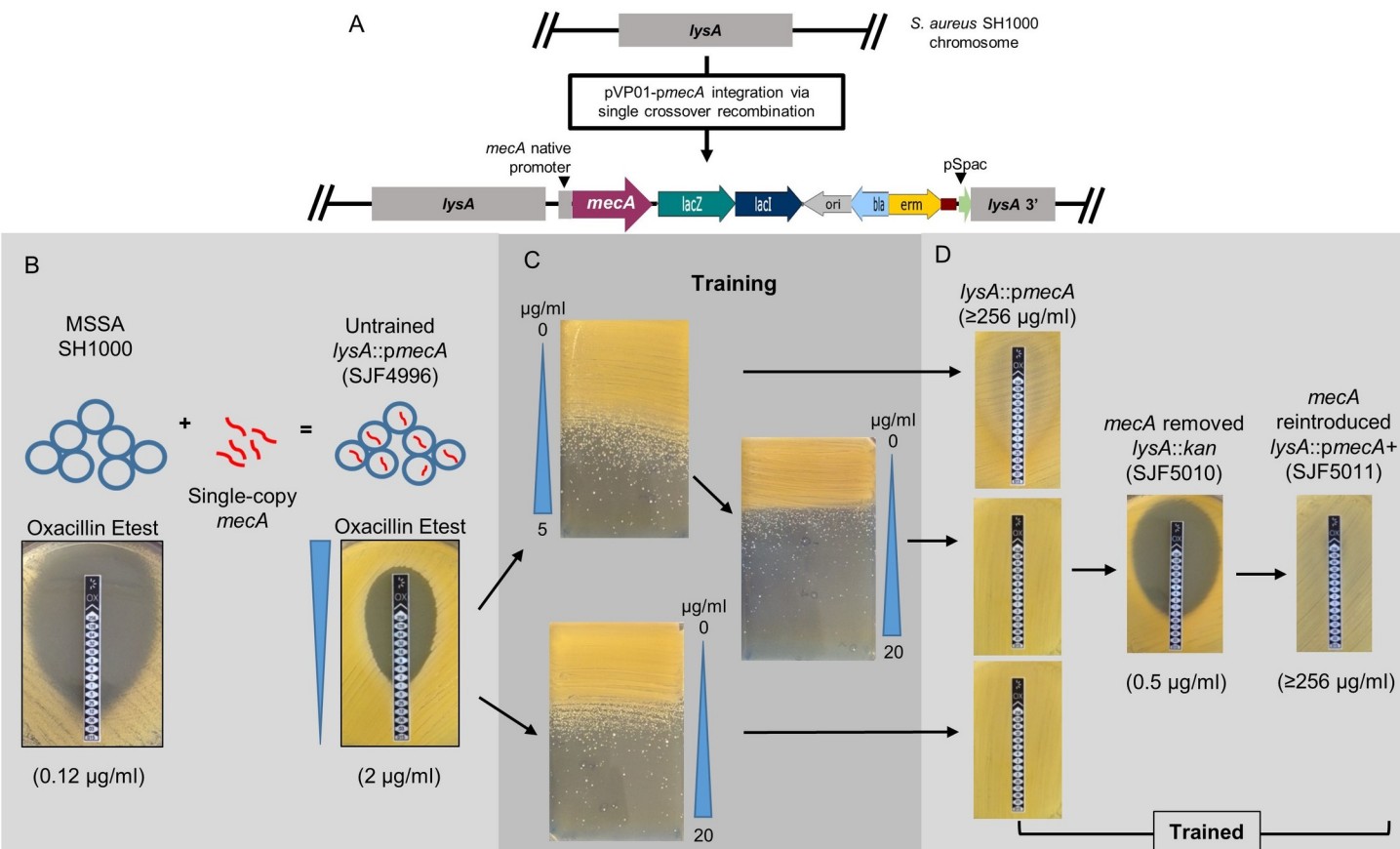

**Fig 1. Construction of a chromosomal p*mecA* fusion in *S. aureus* SH1000. A)** Schematic overview of genomic region of *lysA* in SH1000 and post integration of pVP01-p*mecA* resulting into *lysA*::p*mecA* (SJF4996). **B)** The *lysA*::p*mecA* (SJF4996) construct was produced by transduction from RN4220 *lysA*::p*mecA* (SJF4994). This led to single copy *mecA* at the *lysA* locus expressed *mecA* under its native promoter. **C)** Subsequently, spontaneous oxacillin resistant mutants were selected using a gradient plate containing either 0–5 or 0–20 μg/ml of methicillin. Strains with low-level resistance were further selected using an antibiotic gradient plate containing 0–20 μg/ml methicillin to select high-level resistance. **D)** Etest strips revealed high-level oxacillin resistance was lost by removal of p*mecA* and restored via subsequent reintroduction of p*mecA* at the *lysA* locus. Oxacillin MICs shown in brackets were measured using the Etest strip.

genetic manipulation, no other mutations were observed in SH1000 and *lysA*::p*mecA* (SJF4996). The highly resistant strains harboured point mutations in either *rpoB* (*SAOUHSC_00524*; DNA-directed RNA polymerase subunit β) or *rpoC* (*SAOUHSC_00525*; DNA-directed RNA polymerase subunit β') genes (S2 Table). The positions of the nonsynonymous mutations that were identified in *rpoB* and *rpoC* are depicted on physical maps with corresponding gene regions (Fig 2A).

In addition to *rpo*, some resistant strains had acquired other mutations. Strains *lysA*::p*mecA* (SJF5002) and *lysA*::p*mecA* (SJF5006) had amino acid substitutions in the *SAOUHSC_00269* and *SAOUHSC_00270*, which likely code for components of a Type VII secretion system ([29]; S2 Table). Four out of seven highly resistant strains had a base deletion that resulted in a frameshift in the *SAOUHSC_00591* gene encoding another hypothetical protein (S2 Table). Unlike the *rpoB/C* mutations these mutations were not present in all the highly resistant strains.

For two strains, *lysA*::p*mecA* (SJF5006) and *lysA*::p*mecA rpoB*-H929Q (SJF5003), a 12-bp insertion (genome position 2134372) was present compared to NCTC8325 (S2 Table). This insertion maps to *SAOUHSC_02031* (*rsbU*), coding for the SigB regulatory protein RsbU. *S.*

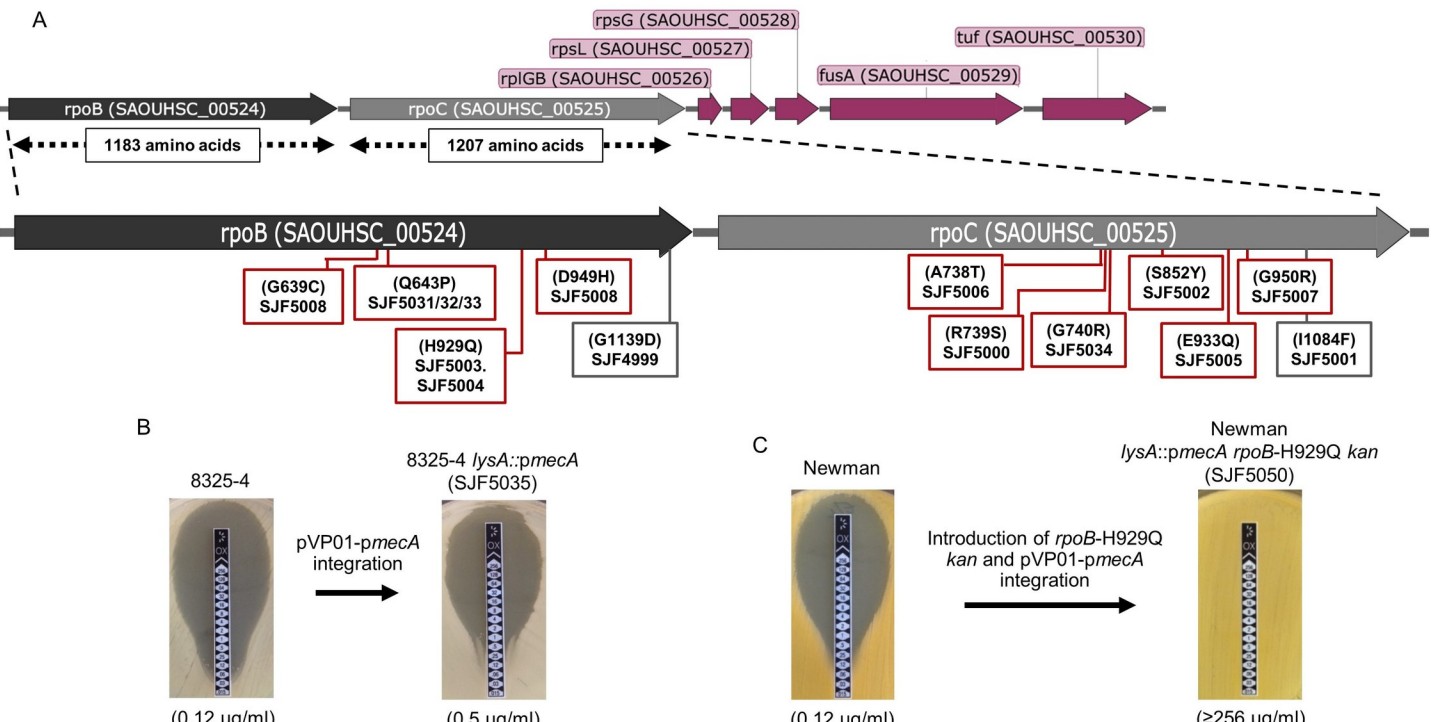

**Fig 2. Identification of *rpoB/C* mutations and reconstitution of 8325–4 and Newman expressing single copy *mecA*. A)** Schematic representation of the genomic region of the *rpoB/C* operon, and downstream genes. Amino acid substitutions identified in resistant derivatives of *lysA*::p*mecA* (SJF4996) are indicated with a red box representing highly resistant mutants with oxacillin MIC of ≥256 μg/ml and a grey box representing intermediate resistant mutants with oxacillin MIC of up to 16 μg/ml. **B)** Chromosomal integration of *mecA* at *lysA* locus in 8325–4 did not result in high-level oxacillin resistance. **C)** The introduction of *lysA*::p*mecA* and *rpoB*-H929Q *kan* of SJF5046 into Newman resulted in the development of high-level resistance. Oxacillin susceptibility of Newman *lysA*::p*mecA rpoB*-H929Q *kan* (SJF5050) and its parental strain Newman were compared using Etest.

*aureus* NCTC8325 carries a deletion in *rsbU*, which is repaired in SH1000 [30]. To investigate the possible effects of SigB dysregulation, a phage lysate from *lysA*::p*mecA* (SJF4994) was transduced into NCTC8325-4, a derivative of NCTC8325 with three prophages removed [31], creating *S. aureus* 8325–4 *lysA*::p*mecA* (SJF5035) with a defective *rsbU* gene. The oxacillin sensitivity of *S. aureus* 8325–4 *lysA*::p*mecA* (SJF5035) (MIC 0.5 μg/ml) was similar to that of the parental strain (MIC 0.12 μg/ml) (Fig 2B), suggesting that *rsbU* is unlikely to play an important role in high level resistance.

Previous studies have reported that alterations in *rpoB* are involved in developing resistance to methicillin, vancomycin and daptomycin [32]. Moreover, mutation in *rpoC* converts heterogeneously vancomycin-intermediate *S. aureus* (hVISA) into slow VISA which results in increased oxacillin resistance [33,34]. All identified *rpoB/C* mutations associated with high-level oxacillin resistance identified here resulted in amino acid substitutions towards the *C*-termini of RpoB and RpoC (Fig 2A). It has previously been reported that the locations of point mutations in the *rpoB* gene determine the resistance to specific antibiotics. For example, RpoB-H481Y is associated with high-level resistance to rifampicin as well as intermediate resistance to vancomycin and RpoB-N967I and RpoB-R644H enhance methicillin, but not rifampicin resistance [21,32,35]. The existence of *rpoB/C* mutations in clinical MRSA isolates related to various lineages and their effect on developing β-lactam resistance was confirmed by sequence alignment of strains, COL, USA300, MRSA252, Mu50 and JH9 (S2 Table). Interestingly, USA300_FPR3757 has an *rpoC* mutation and yet it is quite sensitive to oxacillin. This

could be due to either additional mutation elsewhere in the chromosome or overexpression of regulatory genes that could result in oxacillin susceptibility.

To test if the *rpoB/C* mutations lead to high-level β-lactam resistance, two highly resistant, strains *lysA*::p*mecA rpoB*-H929Q (SJF5003) and *lysA*::p*mecA rpoC*-G740R (SJF5034) were restored to wild-type *rpoB* and *rpoC*, respectively. Bacteriophage mediated transduction of the above strains with an AJ1008 [36] phage lysate resulted in the introduction of a kanamycin resistance marker; genetically linked to the WT *rpoB/C* genes. The resulting strains *lysA*::p*mecA rpoB*⁺ (SJF5044) and *lysA*::p*mecA rpoC*⁺ (SJF5045) were oxacillin sensitive, indicating that *rpoB* or *rpoC* mutations are needed for high-level resistance (S1A Fig). A similar approach was used for *S. aureus* COL, replacing native *rpoB* (A798V, S875L) with SH1000 *rpoB*. The resultant strain COL *rpoB*⁺ (SJF5049), was highly sensitive to oxacillin (MIC 0.5 μg/ml; S1A Fig). Collectively, these observations confirm that the presence of *rpoB/C* mutations is sufficient to develop oxacillin resistance in *mecA* positive strains.

We next addressed whether the *mecA/rpoB/C*-dependent high-level oxacillin resistance is strain specific. Newman [37] was used for complete reconstitution of the high-level resistance phenotype. In order to introduce the H929Q mutation into the *rpoB* gene of *S. aureus* Newman, a phage lysate from AJ1008 with *kanA* near *rpoB/C* [36] was transduced into *lysA*::p*mecA rpoB*-H929Q (SJF5003) and resultant transductants were screened for both kanamycin- and oxacillin-resistance. This resulted in strain *lysA*::p*mecA rpoB*-H929Q *kan* (SJF5046). A phage lysate from SJF5046 was used to transduce the *rpoB*-H929Q mutation into Newman. Subsequently, p*mecA* was introduced at the *lysA* locus resulting in Newman *lysA*::p*mecA rpoB*-H929Q *kan* (SJF5050). This strain (SJF5050) was oxacillin resistant (MIC≥256 μg/ml) compared to Newman (MIC 0.12 μg/ml) (Fig 2C) and demonstrated that *mecA/rpoB/C*-dependent high-level oxacillin resistance is not a strain-specific phenomenon.

## Effects of *rpoB* and *rpoC* mutations on the activity of RNAP

The mutations in RNAP may alter enzyme properties and subsequent gene expression leading to phenotypic conversion to high-level resistance. To test this, we purified RNAPs from wild-type SH1000, *lysA*::p*mecA rpoB*-H929Q (SJF5003) and *lysA*::p*mecA rpoC*-G740R (SJF5034). Transcription elongation can be the rate limiting, and, thus, the main regulatory step of gene expression [38–41]. We therefore tested purified RNAPs for all activities during elongation using artificially assembled elongation complexes [42–44]. The rates of nucleotide addition and pyrophosphorolysis were not, or only slightly, affected by the mutations in *rpoB*-H929Q and *rpoC*-G740R (S2 Fig), suggesting that catalysis of phosphodiester bond formation was not altered. In contrast there were strong, but different, effects of *rpoB*-H929Q and *rpoC*-G740R mutations on pausing at some of the positions of the template (S3 Fig). Transcription pausing is the main way of regulation of transcription elongation, proving a potential link to altered cell phenotype.

Transcription pausing can be caused by misincorporation of nucleotides [39,40]. However, the rates of misincorporation were similar for all three RNA polymerases (S4 Fig). RNA cleavage is used to resolve misincorporation and backtracking events during transcription and the apparent rate constant ($k_{obs}$) for intrinsic cleavage of mis-incorporated nucleotides by the *rpoC*-G740R mutant RNA polymerase was almost an order of magnitude faster than the native RNAP (S5A Fig). This only applied to the intrinsic cleavage activity of *rpoC*-G740R mutant RNAP, because Gre factor-assisted RNA cleavage in the same complex remained the same for all three RNA polymerases (S5B Fig). Thus, *rpoC*-G740R mutation may positively affect expression of the genes that are particularly affected by backtracking or contain misincorporation hotspots [40].

Transcription initiation is also a heavily regulated step of transcription. We reconstructed SH1000, *lysA*::p*mecA rpoB*-H929Q (SJF5003) and *lysA*::p*mecA rpoC*-G740R (SJF5034) RNAP holoenzymes with housekeeping SigA and alternative sigma factor SigB. Their abilities to initiate transcription *in vitro* (synthesis of trinucleotide) on *pbp2, mecA and clfB* promoters recognised by SigA and the *asp23* promoter recognised by SigB were compared. Unlike the above experiments, initiation *in vitro* depends on specific activity of RNAPs. We, therefore, normalised all the results to the activities of RNAPs on a strong orthologous A1 promoter of *E. coli* bacteriophage T7 (*rpoB*-H929Q – 0.65%, *rpoC*-G740R – 0.64% of SH1000 RNAP activity; Fig 3A). The *rpoB*-H929Q mutant RNAP was impaired in initiating transcription from the SigA-dependent promoters compared to the native and *rpoC*-G740R RNAP (Fig 3A). In contrast, all the RNAPs behaved similarly at the SigB-dependent *asp23* promoter (Fig 3A). The results therefore suggest that differential gene expression in the *lysA*::p*mecA rpoC*-G740R strain (SJF5034) is caused mainly by alterations in elongation of transcription, while the *lysA*::p*mecA rpoB*-H929Q strain (SJF5003), by changes both of elongation and initiation of transcription. We also tested whether the oxacillin intervention affect the transcription. RNAPs of SH1000, *lysA*::p*mecA rpoB*-H929Q (SJF5003) and *lysA*::p*mecA rpoC*-G740R (SJF5034) had no effect on transcription in the presence of high and low concentrations of oxacillin (S6 Fig).

## Mutations in RNAP decrease emergence of Rifampicin resistance

The impact of *rpoB/C* mutations on RNAP was examined by mapping *S. aureus rpoB/C* mutations onto the cryoEM structure of the *E. coli* RNAP elongation complex [45]. All *S. aureus* mutations are in conserved residues with *E. coli*, except for one in RpoC (S852). The mutations predominantly mapped either on the surface or towards the surface far from the catalytic center (Fig 3B) [46]. Rifampicin (RMP) is an antibiotic that acts by binding within the RNAP β subunit [47]. None of the identified mutations were located in the regions of binding of RMP [36,47], which coincided with rifampicin sensitivity of the strains (Fig 3C; S2 Table). RMP resistance is easily acquired (frequency of $\sim 10^{-8}$) via point mutations within *rpoB* [48,49], but these also may confer a great fitness cost [50]. We hypothesized that as MRSA strains already carry mutations in *rpoB* or *rpoC*, they may not be able to develop supplementary RMP resistance. As expected, resistant mutants occurred with frequencies of around $10^{-8}$ in SH1000 and *lysA*::p*mecA* (SJF4996) (S5 Table). Conversely, *lysA*::p*mecA rpoC*-G740R (SJF5034) had a resistance frequency of around $10^{-10}$ and for strains having the *rpoB*-G1139D and *rpoB*-Q643P mutations RMP resistance was not detected (frequency $<5\times10^{-11}$; S5 Table). This suggests the potential of RMP and its derivatives to treat infections caused by high level MRSA strains with considerably lower risk of developing RMP resistance.

## Mutations in *rpoB* and *rpoC* contribute to slower growth rates and increased PBP2A production

In the absence of antibiotic, *lysA*::p*mecA* (SJF4996) exhibited a growth rate similar to that of SH1000 (doubling times ~36 and ~35 min, respectively; Fig 4A). However, *lysA*::p*mecA rpoB*-H929Q (SJF5003), *lysA*::p*mecA rpoC*-G740R (SJF5034) and *lysA*::*kan rpoB*-H929Q (SJF5010), exhibited slower growth rates (doubling times ~48, ~55 and ~49 min, respectively) and similar to COL (doubling time ~48 min). In the presence of 20 μg/ml oxacillin, the growth rate of *lysA*::p*mecA rpoB*-H929Q (SJF5003), *lysA*::p*mecA rpoC*-G740R (SJF5034) and COL was unaffected (Fig 4A). These observations suggest that the growth of highly resistant *rpoB/C* mutants remains stable in the presence of oxacillin, similar to the clinical isolate COL. As *lysA*::*kan rpoB*-H929Q (SJF5010) is oxacillin sensitive, slow growth cannot account for increased resistance.

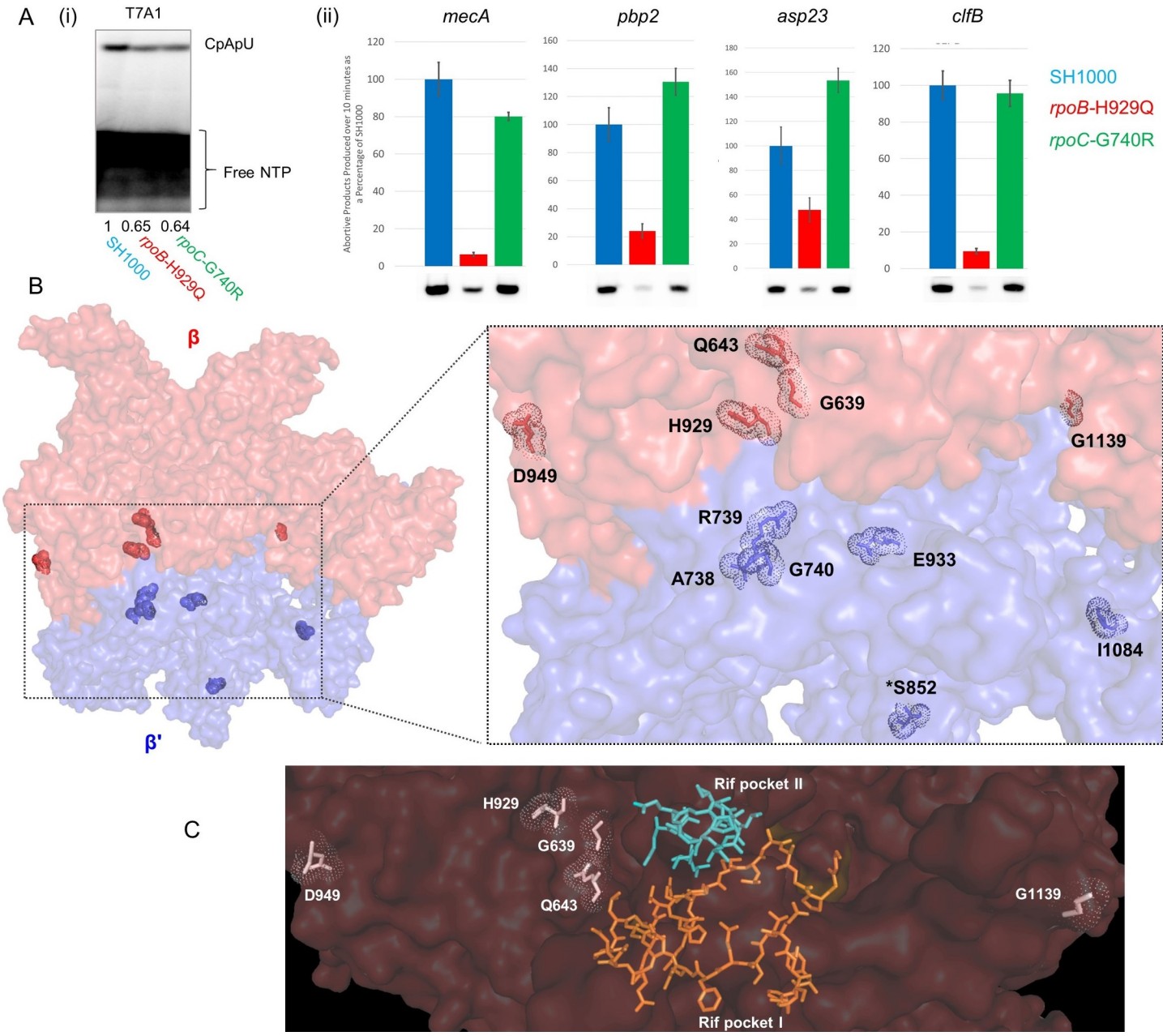

**Fig 3. Location of *rpoB/C* mutations and the impact on transcription initiation. A)** (i) Abortive products produced on the T7A1 promoter by *S. aureus* RNAPs over 10 min at 37˚C. (ii) Abortive initiation by SH1000, *lysA*::p*mecA rpoB*-H929Q (SJF5003) and *lysA*::p*mecA rpoC*-G740R (SJF5034) holoenzymes on the promoters of *mecA*, *pbp2*, *asp23* and *clfB*. Abortive products were normalised to abortive synthesis on T7A1 promoter. Error bars show ±SD from three replicates. **B)** The three-dimensional structure of *E. coli* RNAP holoenzyme was used to map the *rpoB* and *rpoC* mutations [67]. The two subunits β (red) and β' (blue) are shown as molecular surface. The mutated residues of *rpoB/C* (*S. aureus* residue numbering) are highlighted with dots (Zoomed in section). *, residue not conserved in *E. coli*. **C)** *S. aureus rpoB* mutations mapped on the structure of *E. coli* RpoB shows relative location of the mutations to the rifampicin binding pocket regions l and ll [47], shown as orange and cyan sticks, respectively. Figure generated from PDB 6CUX using PYMOL.

In order to determine whether there was any correlation between oxacillin resistance and the level of PBP2A, we compared the single copy chromosomal p*mecA* (*lysA*::p*mecA*) to strains expressing plasmid-borne *mecA* (pRB474-p*mecA*) under the control of its native promoter (S7 Fig; S2 and S4 Tables). Plasmid-borne *mecA* strains were evolved similarly to our single copy

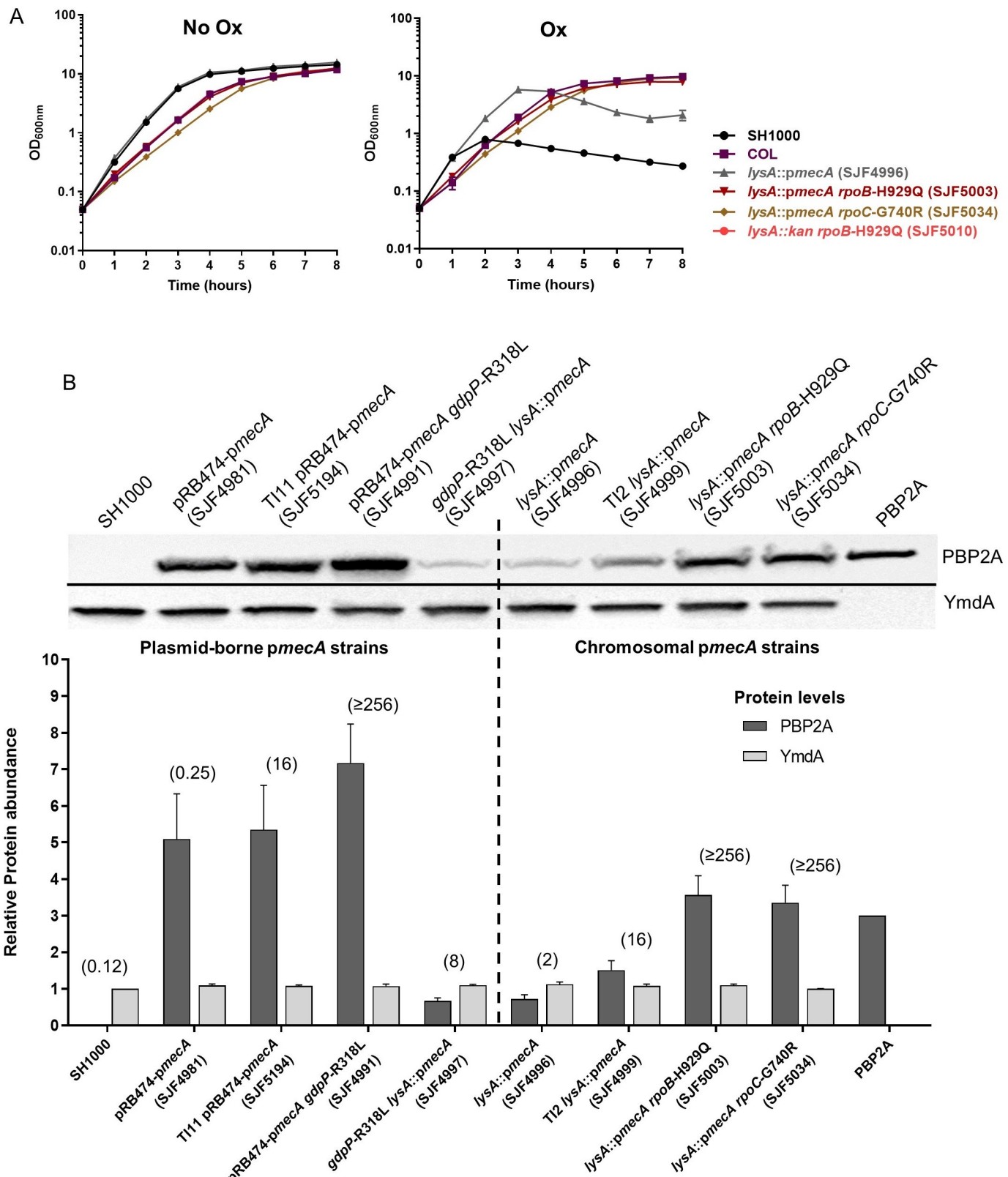

**Fig 4. Growth characteristics and PBP2A production of *S. aureus* strains. A)** Growth of representative strains, *lysA*::p*mecA rpoB*-H929Q (SJF5003), *lysA*::*kan rpoB*-H929Q (SJF5010), *lysA*::p*mecA rpoC*-G740R (SJF5034) were compared to the SH1000, COL and *lysA*::p*mecA* (SJF4996) in the absence (left) and presence

(right) of oxacillin. Following one hour of incubation, 0.5 µg/ml of oxacillin was added to SH1000 and *lysA*::p*mecA* (SJF4996) cultures, and 20 µg/ml of oxacillin was added to COL, *lysA*::p*mecA rpoB*-H929Q (SJF5003) and *lysA*::p*mecA rpoC*-G740R (SJF5034) cultures. Bacterial cultures were prepared in triplicate and the error bars represent standard deviation of the mean (±SEM). **B)** Whole cell lysates (~10 µg of protein) of strains expressing chromosomal p*mecA* (single-copy; *lysA*::p*mecA*) and plasmid-borne p*mecA* (multi-copy; pRB474-p*mecA*) were used to determine the amounts of PBP2A (~76kDa) and were compared between these two set of strains. Plasmid-borne p*mecA* carrying strains include, pRB474-p*mecA* (SJF4981), TI11 pRB474-p*mecA* (SJF5194), pRB474-p*mecA gdpP*-R318L (SJF4991) and *gdpP*-R318L *lysA*::p*mecA* (SJF4997). Chromosomal p*mecA* expressing strains include, *lysA*::p*mecA* (SJF4996), TI2 *lysA*::p*mecA* (SJF4999), *lysA*::p*mecA rpoB*-H929Q (SJF5003) and *lysA*::p*mecA rpoC*-G740R (SJF5034). Anti-YmdA antibodies were used as an endogenous control producing ~58 kDa band. Relative levels of PBP2A were calculated using ImageLab (Bio-Rad) quantitation tool, selecting recombinant PBP2A as a point of reference. The results of relative concentrations of PBP2A are the average of three independent repeats where error bars represent standard deviation of the mean (±SEM). Oxacillin MICs are listed in brackets for all strains above the bars.

strains (S7 Fig). Apart from the genetic manipulation itself, no other mutations were observed in SH1000 and SH1000 pRB474-p*mecA* (SJF4981). Oxacillin resistant derivatives of SH1000 pRB474-p*mecA* (SJF4981) harboured mutations in the *gdpP* gene encoding c-di-AMP phosphodiesterase [24] and not *rpoB/C*. Disruption or deletion of *gdpP* was found to be associated with high-level resistance (S8B Fig and S4 Table). Removal and subsequent reintroduction of pRB474-p*mecA* regained susceptibility and resistance, respectively in strains *gdpP*-R318L (SJF4993) and *gdpP*-R318L pRB474-p*mecA*⁺ (SJF4995). To test if *gdpP* or *rpoB* could lead to high level resistance irrespective of *mecA* copy number, strains *gdpP*-R318L *lysA*::p*mecA* (SJF4997) and *lysA*::*kan rpoB*-H929Q pRB474-p*mecA* (SJF5024) (S9 Fig) were constructed. Interestingly, *gdpP*-R318L *lysA*::p*mecA* (SJF4997) developed only low-level resistance to oxacillin (MIC 8 µg/ml; S9A Fig) and *lysA*::*kan rpoB*-H929Q pRB474-p*mecA* (SJF5024) was sensitive (MIC 0.5 µg/ml; S9B Fig). This suggests there are two independent mechanisms leading to high-level resistance, where the delivery route of *mecA* determines which pathway is utilized. Elucidation of the mechanism underlying this phenomenon requires further study.

Using the single copy and plasmid-borne *mecA* strains with a range of resistance properties the level of PBP2A in the cells was compared (Fig 4B). Strain pRB474-p*mecA gdpP*-R318L (SJF4991) produced the highest levels of PBP2A and was highly resistant, compared to the rest of the strains (Fig 4B). Highly resistant strains with chromosomal p*mecA*; *lysA*::p*mecA rpoB*-H929Q (SJF5003) and *lysA*::p*mecA rpoC*-G740R (SJF5034) showed only two-fold increase in the amounts of PBP2A compared to *lysA*::p*mecA* (SJF4996) (Fig 4B) and less PBP2A than plasmid-borne strains with lower resistance levels.

## Comparative transcript profiling reveals the effect of resistance evolution on gene expression

To determine how resistance to β-lactam antibiotic evolves, the effect of *mecA* on global gene expression was determined. Five *S. aureus* strains were used for transcript profiling: WT (SH1000), *lysA*::p*mecA* (SJF4996), *lysA*::p*mecA rpoB*-H929Q (SJF5003), *lysA*::*kan rpoB*-H929Q (SJF5010) and *lysA*::p*mecA rpoC*-G740R (SJF5034) (S2 Table). Differentially expressed genes (DEGs) were identified by pair-wise and group-wise comparisons. Genes with an adjusted *p* value <0.05 and log₂ fold-change ≥±1 were deemed to be statistically significant.

### (i) Determination of altered gene expression in response to acquisition of *mecA* and *rpo* mutations

Comparing *lysA*::p*mecA* (SJF4996) to SH1000 revealed 193 DEGs. In contrast, only 9 DEGs were identified for the *lysA*::p*mecA rpoB*-H929Q (SJF5003) compared to SH1000 (Fig 5A), suggesting that expression of 190 genes had reverted to WT levels (Fig 5A). A total of 120 genes were differentially expressed in *lysA*::p*mecA rpoC*-G740R (SJF5034) compared to SH1000, meaning that 73 genes were back at WT levels. One hundred and twenty two DEGs

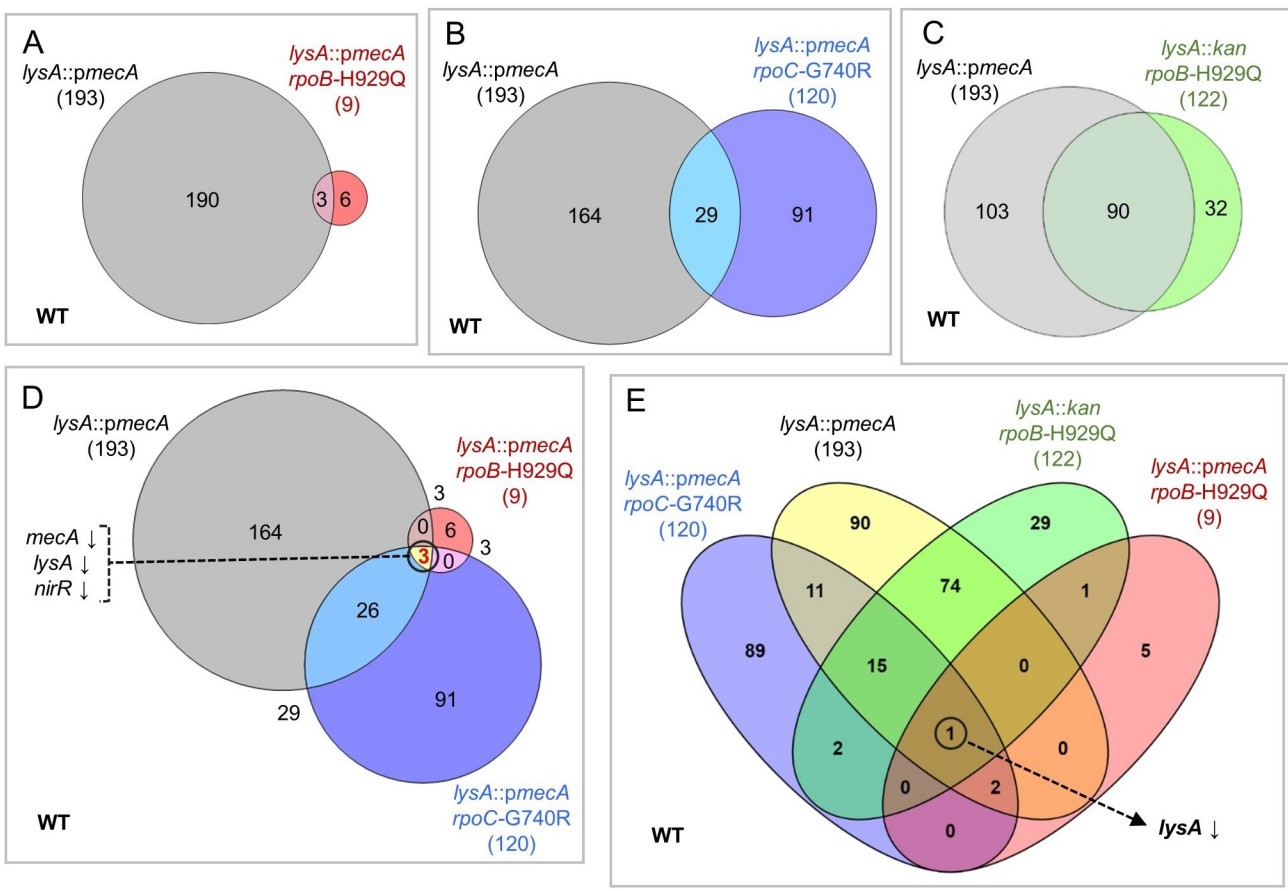

**Fig 5. Differential gene expression in SH1000 follows the acquisition of chromosomal *mecA* and the *rpoB/C* mutations.** Two-set proportional Venn diagram illustrating the total number of differentially expressed genes (DEGs) altered in A) *lysA*::p*mecA* (SJF4996) (grey) and *lysA*::p*mecA rpoB*-H929Q (SJF5003) (red), B) *lysA*::p*mecA* (SJF4996) (grey) and *lysA*::p*mecA rpoC*-G740R (SJF5034) (blue) and C) *lysA*::p*mecA* (SJF4996) (grey) and *lysA*::*kan rpoB*-H929Q (SJF5010) (green) compared to WT (white background). D) Three-set proportional Venn diagram displaying the shared DEGs among *lysA*::p*mecA* (SJF4996) (grey), *lysA*::p*mecA rpoB*-H929Q (SJF5003) (red) and *lysA*::p*mecA rpoC*-G740R (SJF5034) (blue) compared to WT (white background). 3 common DEGs (*mecA*, *lysA*, *nirR*) are marked by a circle showing reduced expression. E) The *lysA* gene (marked by a circle) was shared among four strains based on comparisons from *c*) and *d*) compared to WT (white background). The expression of *lysA* was reduced in all four strains compared WT, marked by a downward arrow.

were solely associated with the *rpoB*-H929Q mutation of *lysA*::*kan rpoB*-H929Q (SJF5010) compared to SH1000 (Fig 5C). These data illustrate the profound transcriptional changes associated with the presence of *mecA* and how many of these changes are reversed upon subsequent acquisition of *rpoB* or *rpoC* mutations in the highly resistant strains.

Group-wise comparison of *lysA*::p*mecA* (SJF4996), *lysA*::p*mecA rpoB*-H929Q (SJF5003) and *lysA*::p*mecA rpoC*-G740R (SJF5034) to SH1000 revealed the presence of only three common DEGs. These were *lysA*, *mecA* and *nirR* (Fig 5D). The presence of *lysA* and *mecA* in this group was not surprising, as the *lysA*::p*mecA* (SJF4996) was created by inserting the *mecA* gene at the 3' region of the *lysA* gene. Upregulation of *nirR* (Nitrate reductase regulator) was apparent in *lysA*::p*mecA* (SJF4996) whereas, in *lysA*::p*mecA rpoB*-H929Q (SJF5003) and *lysA*::p*mecA rpoC*-G740R (SJF5035), it was downregulated, suggesting that the presence of *mecA* alone induces the expression of *nirR* (Fig 5D). Only *lysA* was found to be common upon addition of *lysA*::*kan rpoB*-H929Q (SJF5010) into group-wise comparison relative to WT (Fig 5E).

Hence, the *rpoB/C* mutations, particularly *rpoB*-H929Q, appear to dampen the perturbations in gene expression caused by acquisition of *mecA*, whilst permitting enhanced *mecA* expression, suggesting that 'normalising' global gene expression is a requirement for the development of *mecA*-dependent high-level β-lactam resistance.

## (ii) Differentially expressed genes associated with high-level resistance

Strains *lysA*::p*mecA rpoB*-H929Q (SJF5003) and *lysA*::p*mecA rpoC*-G740R (SJF5034) had 121 common DEGs compared to *lysA*::p*mecA* (SJF4996) (Fig 6A and S6 Table). Strains *lysA*::p*mecA rpoC*-G740R (SJF5034) and *lysA*::*kan rpoB*-H929Q (SJF5010) had 35 common DEGs relative to *lysA*::p*mecA* (SJF4996) (Fig 6B). Two pair-wise comparisons of *lysA*::p*mecA rpoB*-H929Q (SJF5003) and *lysA*::*kan rpoB*-H929Q (SJF5010) strains against *lysA*::p*mecA* (SJF4996) were combined to determine the transcriptional changes resulting from the *rpoB*-H929Q mutation alone, revealing 28 common DEGs (Fig 6C). Finally, strains *lysA*::p*mecA rpoB*-H929Q (SJF5003), *lysA*::*kan rpoB*-H929Q (SJF5010) and *lysA*::p*mecA rpoC*-G740R (SJF5034); were compared to *lysA*::p*mecA* (SJF4996) yielding, 26 DEGs common to all three *rpo* strains. Notably, 22 DEGs were up-regulated in the *lysA*::p*mecA* (SJF4996) strain whereas, the same 22 genes were down-regulated in the three *rpo* strains (Fig 6D(i) and 6(ii)).

Functional categorisation of the 121 DEGs (95 specifically associated with high-level resistance in response to the acquisition of *rpoB*-H929Q and *rpoC*-G740R plus 26 with altered

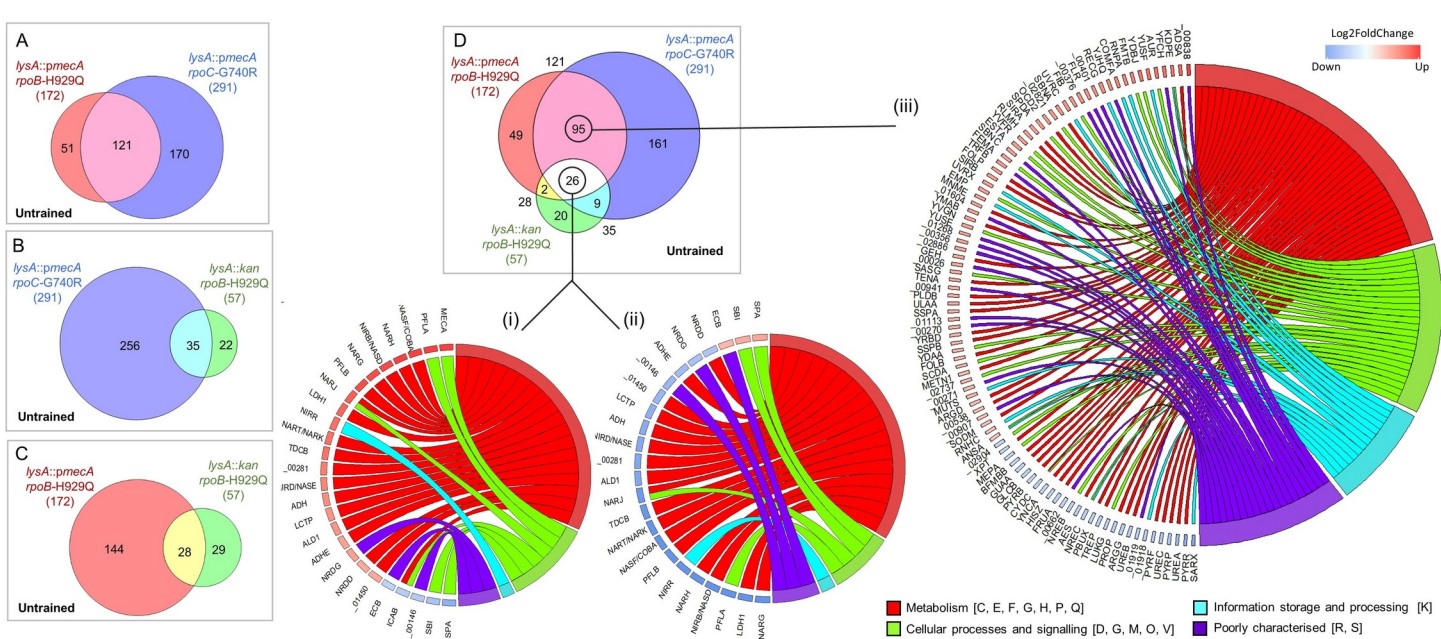

**Fig 6. Identification and functional characterisation of overlapping genes and their differential expression in derived strains with a range of resistance properties.** Two-set proportional Venn diagrams displaying the total number of DEGs (listed in brackets), common (intersection) and unique DEGs altered in **A)** *lysA*::p*mecA rpoB*-H929Q (SJF5003) (red) and *lysA*::p*mecA rpoC*-G740R (SJF5034) (blue), **B)** *lysA*::p*mecA rpoC*-G740R (SJF5034) (blue) and *lysA*::*kan rpoB*-H929Q (SJF5010) (green) and **C)** *lysA*::p*mecA rpoB*-H929Q (SJF5003) (red) and *lysA*::*kan rpoB*-H929Q (SJF5010) compared to *lysA*::p*mecA* (SJF4996) (white background). **D)** Three-set proportional Venn diagram comparing the three *rpo* strains relative to the *lysA*::p*mecA* (SJF4996) illustrated the 26 common DEGs among all three *rpo* strains and 95 overlapping DEGs exclusively common between *lysA*::p*mecA rpoB*-H929Q (SJF5003) (red) and *lysA*::p*mecA rpoC*-G740R (SJF5034) (blue). Chord plots illustrating the differential expression (DE) of 26 common genes (i) from *lysA*::p*mecA* (SJF4996) compared to WT. 24 out of 26 common genes (ii) were differentially expressed in the three *rpo* strains compared to *lysA*::p*mecA* (SJF4996). *mecA* was not included as *lysA*::*kan rpoB*-H929Q (SJF5010) showed an obvious reduced *mecA* expression. Also, *icaB* was not included because it was upregulated only *lysA*::p*mecA rpoB*-H929Q (SJF5003). The DE and functional categorisation of 95 common genes (iii) between *lysA*::p*mecA rpoB*-H929Q (SJF5003) and *lysA*::p*mecA rpoC*-G740R (SJF5034) compared to *lysA*::p*mecA* (SJF4996) strain. Each segment on the left represents the average log2FoldChange of individual genes from the three strains (SJF5003, SJF5010 and SJF5034) which connects to their functional categories on the right.

expression in all three *rpo* strains) was achieved using the Clusters of Orthologous Groups (COGs) database [51] (S6 Table).

Regulation of the 26 DEGs that were common to all three strains (SJF5003, SJF5010 and SJF5034) was largely accounted for by one regulator, Rex (redox-sensing transcriptional repressor; 16 genes) (S6 Table and Fig 7B). Rex responds to changes in the intracellular NADH:NAD$^+$ ratio to regulate genes associated with anaerobic metabolism [52]. In fact, 22 of 26 DEGs common to the three *rpo* strains (compared to the *lysA*::p*mecA* (SJF4996) strain) were linked to anaerobic metabolism. The intracellular NADH:NAD$^+$ ratio increases when there is a ready supply of nutrients but aerobic respiration is impaired, usually due to oxygen limitation. This change is sensed by Rex resulting in the de-repression of Rex-regulated genes [52,53]. Hence genes associated with pyruvate- (*ldh1*, *lctP*, *ald1*, *adhE*, *adh*, *pflA* and *pflB*) and anaerobic respiratory- metabolism (*nirR*, *nirB*, *nirD*, *nasF*, *narG*, *narH*, *narT* and *narJ*), as well as genes coding for the oxygen-independent ribonucleotide reductase (*nrdD* and *nrdG*), were more highly expressed in the *lysA*::p*mecA* (SJF4996) strain but were lowered to SH1000 levels in the three *rpo* strains. Consistent with this interpretation, oxygen consumption rates (OCR) for the *lysA*::p*mecA* (SJF4996) strain were significantly impaired compared to *lysA*::p*mecA* *rpoB*-H929Q (SJF5003) and *lysA*::p*mecA* *rpoC*-G740R (SJF5034) strains (Fig 8). However, the *lysA*::*kan* *rpoB*-H929Q (SJF5010) strain showed similar OCR to the *lysA*::p*mecA* (SJF4996) strain, suggesting *mecA*- dependent and independent effects on cellular respiration. Anaerobic metabolism results in the excretion of organic acids, such as lactate, which leads to culture acidification [54,55]. Increased expression of the urease operon (*ureABC*) in *lysA*::p*mecA* (SJF4996) but not in the three *rpo* strains, is likely a response to culture acidification. These data suggest that expression of *mecA* may impair aerobic respiration resulting in increased intracellular NADH:NAD$^+$ ratios and de-repression of Rex-regulated 'anaerobic' genes, even in the presence of oxygen.

## Functional analysis of genes in high level resistance

The transcriptional profiling revealed several genes as candidate genes with potential roles in endowing high level *mecA*-dependent resistance. These included genes involved in anaerobic respiration and the associated regulatory gene *rex*. Also 11 genes were up-regulated in the *rpo* mutant strains but not in *lysA*::p*mecA* (SJF4996) (S6 Table), suggesting a role in high level resistance. Mutations were created by a mixture of unmarked deletions and use of available transposon insertions. In order to accommodate the transposon resistance marker a new background strain was constructed (*geh*::p*mecA lysA*::*tet rpoB*-H929Q (SJF5323) (Oxacillin MIC ≥256 μg/ml); S7 Table). The role of 14 genes (*SAHOUHSC_00270*, *00271*, *00841*, *00907*, *00936*, *01113*, *01311*, *02276*, *02331*, *02886*, *01450*, *rex*, *adhE* and *pflB*) in resistance was tested by determination of the effect on oxacillin MIC of the respective mutations in *geh*::p*mecA lysA*::*tet rpoB*-H929Q (SJF5323) (S7 Table). Only the unmarked deletion of *SAOUHSC_00271* (*lysA*::p*mecA rpoB*-H929Q Δ*SAOUHSC_00271* (SJF5344)) exhibited decreased resistance with an oxacillin MIC of 32 μg/ml.

## Effect of high-level resistance on *S. aureus* virulence

The set of strains with defined resistance properties, in isogenic backgrounds and with sequenced genomes provided an ideal platform to establish any correlation between antibiotic resistance and *in vivo* fitness (Fig 9). Groups of mice (*n* = 10) were infected with an individual strain (1 x 10$^7$ CFU) of SH1000, *lysA*::p*mecA* (SJF4996), *lysA*::p*mecA rpoB*-H929Q (SJF5003), *lysA*::*kan rpoB*-H929Q (SJF5010), *lysA*::p*mecA rpoC*-G740R (SJF5034), COL and COL *rpoB*$^+$ (SJF5049). The presence of *mecA* led to increased weight loss and kidney CFU compared to

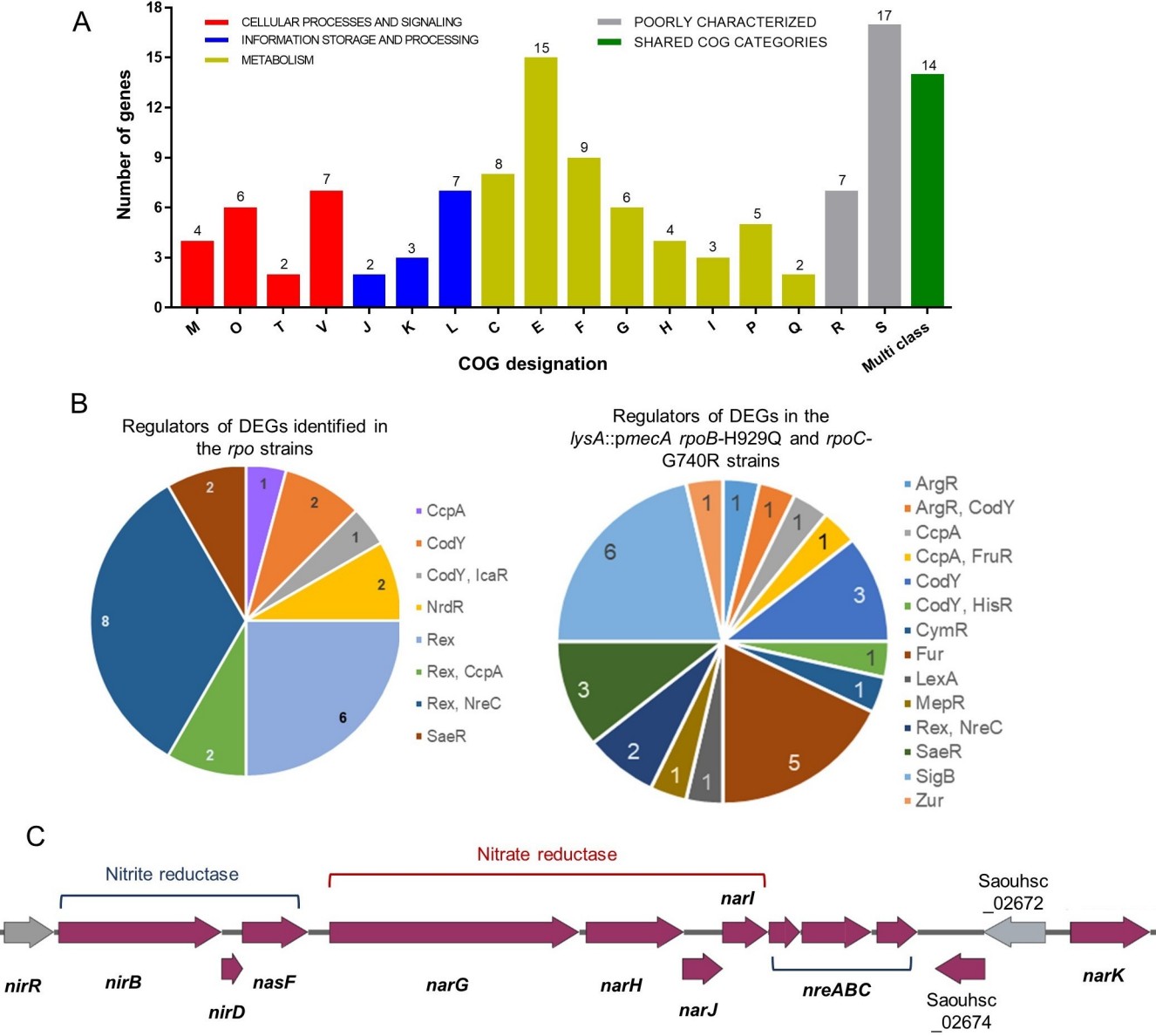

**Fig 7. Functional classification and transcriptional regulation of common genes. A)** COG (Clusters of Orthologous Groups) classification of 121 overlapping DEGs. Colours illustrate COG functional categories. C, Energy production and conversion; E, amino acid transport and metabolism; F, nucleotide transport and metabolism; G, carbohydrate transport and metabolism; H, coenzyme transport and metabolism; I, lipid transport and metabolism; P, inorganic ion transport and metabolism; Q, secondary metabolites biosynthesis, transport, and catabolism; M, cell wall/membrane/envelope biogenesis; O, post-translational modification, protein turnover, and chaperones; T, signal transduction mechanisms; V, defence mechanisms; J, translation, ribosomal structure and biogenesis; K, transcription; L, replication, recombination and repair; R and S are function prediction only or function unknown categories. Total number of genes associated with each category are listed outside the bar. **B)** The transcriptional regulators were predicted for 24 out of 26 and 95 common DEGs identified in the three *rpo* strains compared to their parent (Fig 6D). **C)** Chromosomal region of the genes involved in nitrite and nitrate reduction in *S. aureus*. A membrane bound nitrate reductase system (*narGHJI*) reduces nitrate to nitrite. Subsequently, nitrite is reduced to ammonia by NADH-dependent nitrite reductase system (*nirBD*) [53]. NreABC has been identified as an oxygen sensing system which regulates the expression of nitrate and nitrite reductase system as well as *narK* [68].

SH1000 (Fig 9A–9C). The presence of the *rpo* mutations led to a decrease in weight loss compared to SH1000 (Fig 9A) but no alteration to liver or kidney CFUs (Fig 9B and 9C). In the COL background there was no alteration to pathogenesis associated with the *rpoB* allele (Fig

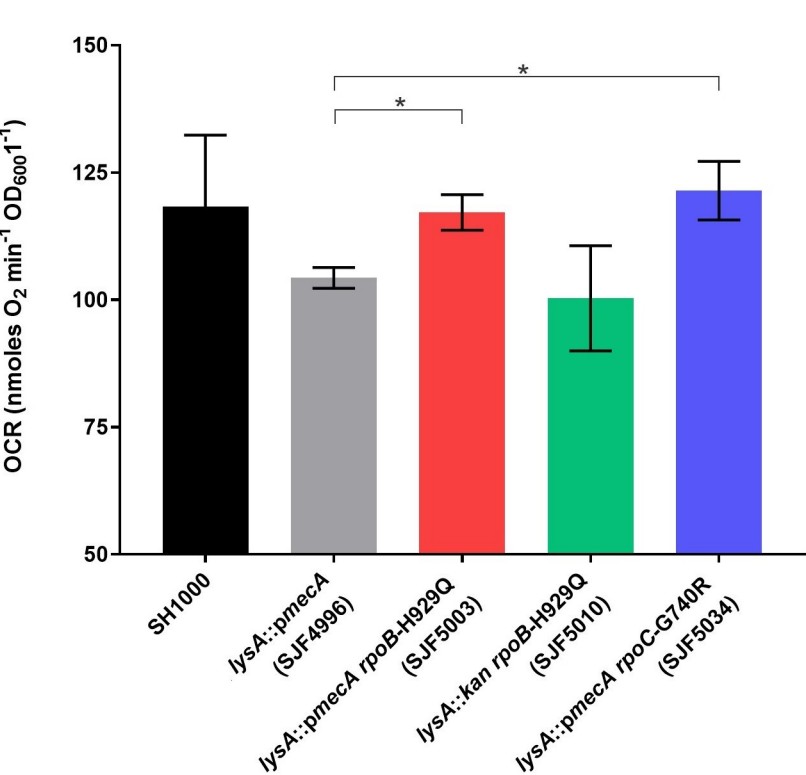

**Fig 8. Oxygen consumption rates of *S. aureus* strains.** Cellular respiration was measured for SH1000, *lysA*::p*mecA* (SJF4996), *lysA*::p*mecA rpoB*-H929Q (SJF5003), *lysA*::*kan rpoB*-H929Q (SJF5010) and *lysA*::p*mecA rpoC*-G740R (SJF5034) strains using Clark-type oxygen electrode. Data shown here reflect mean ±SEM of the three biological replicates. The data were compared and analysed against WT and *lysA*::p*mecA* (SJF4996) using unpaired t-test. * *p* <0.05.

9). Thus, high level resistance is not associated with a significant overall *in vivo* fitness cost, but MecA alone may be advantageous during pathogenesis.

## Discussion

Several studies have used model systems based on expression of *mecA* from multicopy plasmids in MSSA strains to plot the transition from low-level to high-level methicillin resistance [16,20,56]. To match naturally occurring MRSA isolates we introduced a single copy *mecA* into the chromosome of an MSSA strain (SH1000). This strain exhibited low-level resistance and was characterized by enhanced expression of genes involved in anaerobic metabolism, most of which were controlled by Rex (Fig 10). This implies that the redox balance of *S. aureus* SH1000 carrying a chromosomal copy of *mecA* is perturbed with corresponding reduction in aerobic respiration rate.

Selection for high-level resistant derivatives yielded strains with amino acid substitutions either *rpoB* or *rpoC* that largely restored the parental (*S. aureus* SH1000) gene expression profile, i.e. Rex-regulated 'anaerobic' genes were repressed, aerobic respiration rates increased and by implication redox balance was restored (Fig 10). Mutations in *rpoB/C* have been identified previously in highly resistant clinical isolates [21,22,57,58] but have not been well characterised as genetic determinant(s) for developing high-level β-lactam resistance. Interestingly, 11 genes, including *mecA*, were up-regulated in the *rpoB/C* strains exhibiting high-level resistance

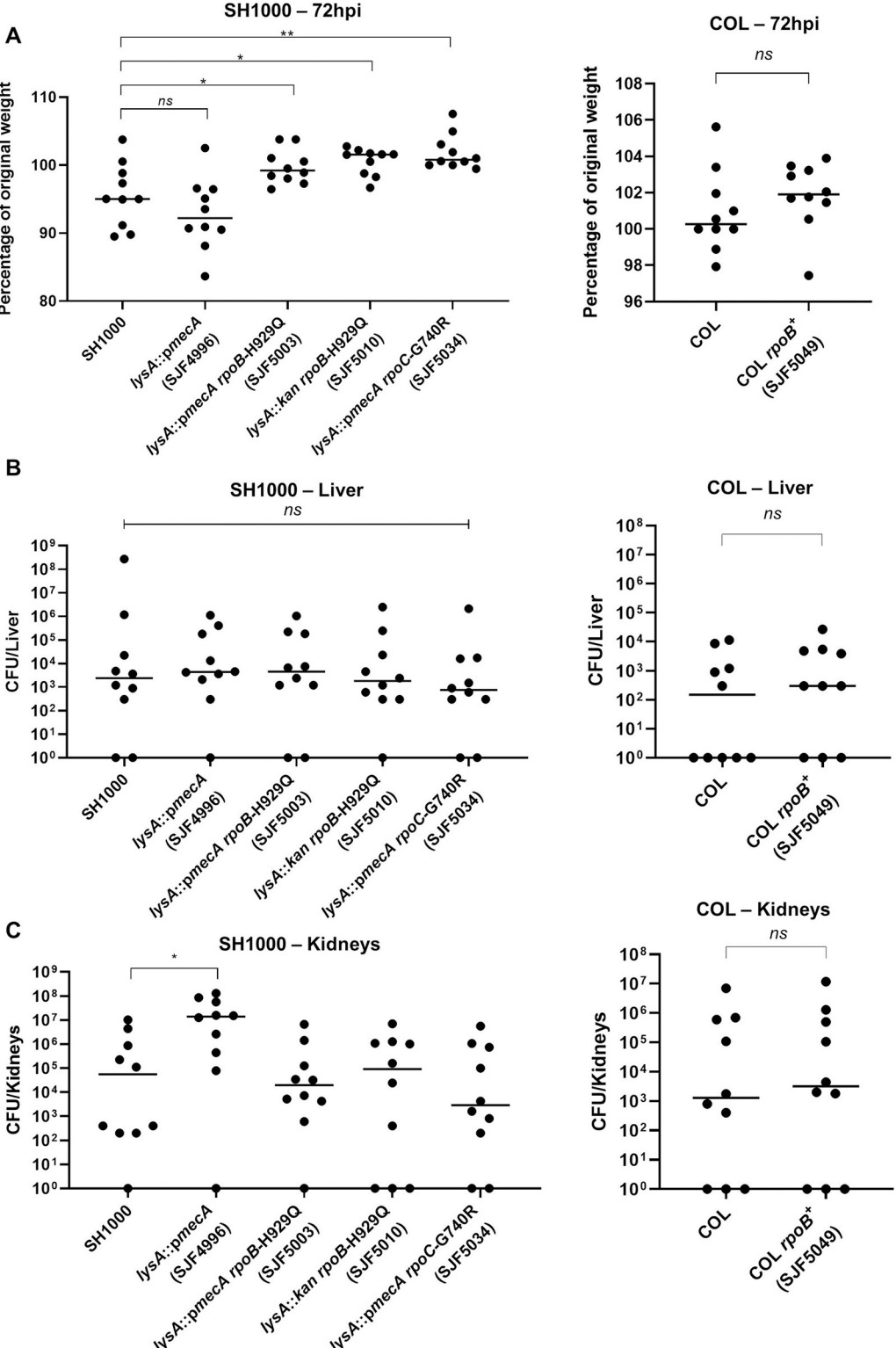

**Fig 9. Virulence of *S. aureus* strains in a murine sepsis model.** Mice were injected with 1 x 10⁷ CFU per mouse with SH1000, *lysA*::p*mecA* (SJF4996), *lysA*::p*mecA rpoB*-H929Q (SJF5003), *lysA*::*kan rpoB*-H929Q (SJF5010), *lysA*::p*mecA rpoC*-G740R (SJF5034), COL, COL *rpoB*⁺ (SJF5049) and animals (*n* = 10 mice per group) were euthanised at 72 hpi. **A)** Weight analysis. Colony forming units (CFUs) per organ **B)** Kidneys and **C)** Livers were recovered from animals infected with strains listed above. Statistical significance determined by Mann-Whitney two-tailed test; *ns*, *p* >0.05; * *p* <0.05; ** *p* <0.005.

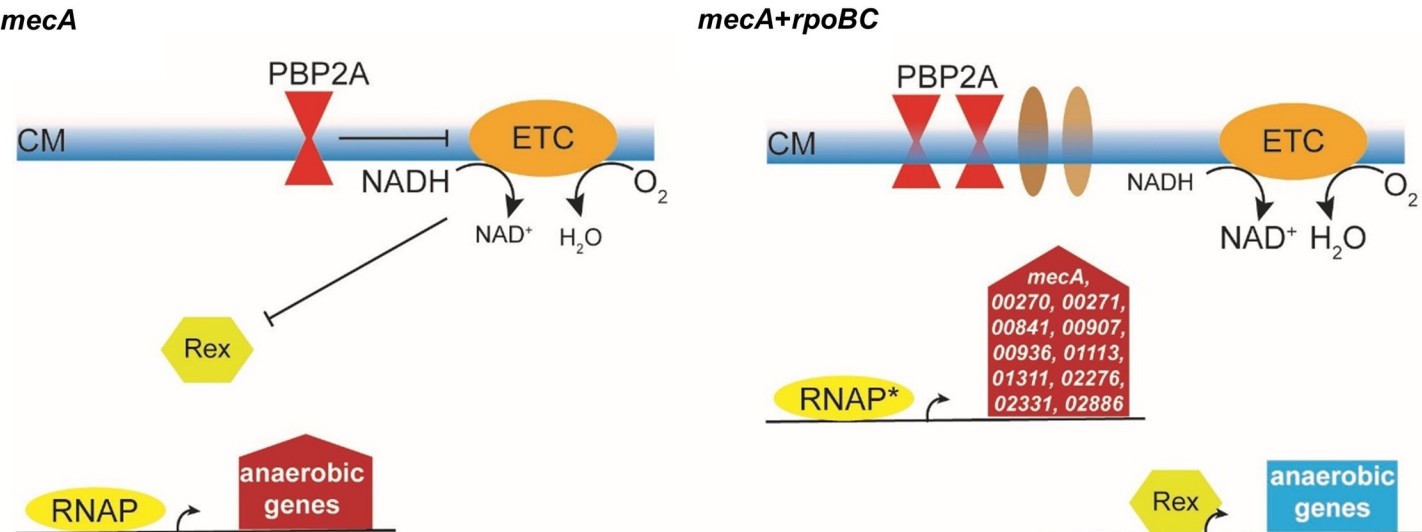

**Fig 10. Model of adaptations associated with the development of high-level β-lactam resistance.** Strain *lysA*::p*mecA* (SJF4996) (left) expresses chromosomal *mecA*, with PBP2A protein (red hour glass) incorporated into the cell envelope. This perturbs cell membrane (CM) function (⊥), resulting in decreased flux through the aerobic electron transport chain (ETC). The resulting increased NADH:NAD$^+$ ratio (indicated by font size) inhibits DNA-binding by the transcription factor Rex (green hexagon) and the anaerobic respiratory and fermentative genes (anaerobic genes) are transcribed (red filled arrow) by the native RNA polymerase (RNAP). This results in a low-level resistance phenotype. Evolved strains have missense mutations in the *rpoB/C* (RNAP*) that permit enhanced expression of 11 genes, including *mecA* and two genes associated with the Type VII secretion system (*SAHOUHSC_00270 and 00271*). Incorporation of some of the gene products (e.g. *SAHOUHSC_00270* and *00271*; brown ellipses) into the cell membrane allows integration of PBP2A activity into 'normal' membrane function. This results in lower NADH:NAD$^+$ ratios, permitting Rex to repress expression of anaerobic genes (blue rectangle), the restoration of aerobic respiratory growth and high level β-lactam resistance.

compared to *lysA*::p*mecA* (SJF4996). The ~2-fold increase in *mecA* expression was matched by a ~2-fold increase in PBP2A protein, which could promote enhanced resistance. However, there was no direct correlation between resistance and PBP2A levels. A functional analysis of the other upregulated genes revealed *SAHOUHSC_00271* as being required for high level resistance. *SAHOUHSC_00270* and *00271*, code for two small membrane proteins of unknown function and are found in the highly variable module 3 region of the Type VII secretion system locus [59]. Interestingly, two high-level resistant strains (SJF5002 and SJF5006) had missense mutations in *SAHOUHSC_00270* and *00269* (S2 Table) [59]. This alludes to a function of the Type VII secretion system in cell envelope function and hence the ability of PBP2A to impart high level resistance (Fig 10).

A key remaining question is how the *rpoB/C* mutations result in increased expression of *mecA* and the other 10 genes shown in Fig 10. A detailed analysis of the properties of *rpoB*-H929Q (SJF5003) and *rpoC*-G740R (SJF5034) RNA polymerases revealed that both exhibited different pausing properties. Pausing plays a major role in regulation of transcription, suggesting that altered pausing behaviour at specific genes could alter their expression [40]. Our defined system will permit the future elucidation of the complex interplay between subtle changes to RNA polymerase and how this manifests as alterations in resistance properties. Acquisition of an alternative component (PBP2A) to maintain the fundamental process of peptidoglycan synthesis in the presence of oxacillin is unlikely to result in optimal performance without further adaptations to integrate the new activity into the cell's physiological and regulatory networks. This is perhaps especially true in the case here, as peptidoglycan synthesis is intimately connected to cell division and membrane function. On the basis of the available evidence, acquisition and expression of *mecA* elicits low-level resistance and perturbs cellular redox balance, resulting in depression of Rex-regulated genes, implying that membrane function is compromised. The missense mutations in *rpoB/C* identified here represent a

route to improved cell performance and high-level resistance. The enhanced transcriptional pausing of these RNAP variants might enhance expression of genes, such as *SAHOUHSC_00271*, to permit PBP2A activity to be integrated into a properly functioning cell envelope, resulting in high-level resistance. Enhanced transcriptional pausing might impair the formation of repressive complexes at those genes specifically up-regulated in the *lysA*::p*mecA rpoB*-H929Q (SJF5003) and *lysA*::p*mecA rpoC*-G740R (SJF5034) strains.

Prior studies have noted that high level resistance is associated with a fitness burden in clinical MRSA limiting the prevalence of these strains [60]. Interestingly, *in vivo* studies indicated that the expression of *mecA* alone leads to slightly higher morbidity making the strain more pathogenic (Fig 9). However, the acquisition of *rpoB* or *rpoC* mutations ameliorates the *mecA* associated effects. Using our defined strains in both the SH1000 and COL backgrounds *rpo* mutations, whilst leading to a reduced growth rate *in vitro*, did not result in a corresponding attenuation *in vivo*. Thus, in the complex interaction with the host high level resistance may not disadvantage the pathogen during infection. This complexity could be compounded by the presence of β-lactamase and the range of *mec* regulatory system variants in the presence of antibiotics.

In conclusion, this study has revealed a novel systemic effect of the acquisition of *mecA* on cellular physiology and how compensatory mutations result in high level antibiotic resistance. Our observations pave the way to begin to elucidate how such an important pathogen *S. aureus* is able to develop high-level drug resistance and provides possible new avenues for therapeutic development.

## Materials and methods

### Bacterial strains, plasmids and growth conditions

The bacterial strains and plasmids used in this study are described in S7 Table. *Escherichia coli* strains were grown in Luria-Bertani (LB) liquid or solid medium supplemented with ampicillin (100 μg/ml) at 37˚C. *S. aureus* strains were grown at 37˚C in Brain-Heart Infusion (BHI) (Oxoid) solid or liquid medium supplemented when required with erythromycin (5 μg/ml) plus lincomycin (25 μg/ml), chloramphenicol (10 μg/ml), tetracycline (5 μg/ml), kanamycin (50 μg/ml), oxacillin (0–20 μg/ml) or methicillin (0–20 μg/ml).

### Construction of strains and plasmids

For chromosomal integration of *mecA*, pMUTIN4 [25] based pGM068 [26] lysine insertion plasmid was used to integrate *mecA* under its native promoter (p*mecA*) downstream of the *lysA* gene. After construction by PCR and restriction/ligation methods, resulting pVP01-p*mecA* plasmids were introduced into *S. aureus* RN4220 by electroporation enabling the integration of plasmid into chromosome via homologous recombination. Oligonucleotides are listed in S8 Table. The resultant erythromycin/lincomycin resistant constructs were then transferred into other *S. aureus* strains as required by Φ11 transduction. For the complementation of *rpoB*-H929Q and *rpoC*-G740R mutations, a phage lysate from AJ1008 [36] with a kan insertion next to *rpoB/C* was used to transduce resistance marker into mutant *S. aureus* strains. Kanamycin resistant but oxacillin sensitive transductants were confirmed by PCR-based assays for functional complementation. SJF5323 (*geh*::p*mecA lysA*::*tet rpoB*-H929Q) was constructed by phage transduction of *lysA*::*tet* and *geh*::p*mecA*-kan into *lysA*::*kan rpoB*-H929Q (SJF5010). *S. aureus* transposon mutants were obtained from the NARSA library (http://www.narsa.net). The transposons from selected strains were transduced into *geh*::p*mecA lysA*::*tet rpoB*-H929Q (SJF5323) (S7 Table) to allow their effects on oxacillin resistance to be determined.

Genomic DNA was isolated using Qiagen DNeasy Blood and Tissue kit (Cat no. 69506) in accordance with manufacturer's instructions. Prior to DNA extraction 5–10 μl of 5 mg/ml lysostaphin was added to resuspended *S. aureus* cells to facilitate lysis. Plasmid isolation was carried out using GeneJET plasmid miniprep kit (Thermo Scientific) as per manufacturer's guidelines.

### Isolation and whole genome sequencing of oxacillin resistant derivatives of *S. aureus* SH1000

*S. aureus* SH1000 *lysA*::p*mecA* (single-copy) and SH1000 introduced with pRB474-p*mecA* (multicopy) plasmid were tested for sensitivity to oxacillin. MIC values were determined using antibiotic susceptibility tests using Etest M.I.C. Evaluator (Oxoid) strips. To obtain high-level resistance to oxacillin, a strain obtained from both approaches (single-copy/multicopy *mecA*) was grown on a methicillin gradient plate where, the bottom layer supplemented with 0-5/ 20 μg/ml methicillin. This enabled the selection of single-copy/multicopy *mecA* derivatives expressing range of oxacillin resistance (2 to ≥256 μg/ml). For the removal of single copy chromosomal *mecA*, pGM068 (Kan$^R$) was phage transduced into oxacillin resistant derivative swapping *lysA*::p*mecA* to *lysA*::*kan*. Curing the pRB474-p*mecA* plasmid from highly resistant strain was achieved by subculturing the strain thirty times in the absence of chloramphenicol.

The genomes of representative strains listed in S7 Table were sequenced by MicrobesNG, University of Birmingham. Genomic DNA was extracted by washing bead stocks with extraction buffer containing lysostaphin and RNase A followed by incubation at 37˚C for 25 min. Proteinase K and RNase A were added and further incubated at 65˚C for 5 min. To purify genomic DNA, equal volume of SPRI (Solid Phase Reversible Immobilisation) beads resuspended in EB buffer was used. DNA was quantified in triplicate using Quantit dsDNA HS assay in a plate reader (Eppendorf). Genomic DNA libraries were prepared using Nextera XT Library Prep Kit (Illumina) in accordance with manufacturer's instructions. Library preparation and DNA quantification was carried out on a liquid handling system (Hamilton Microlab STAR). Pooled libraries were quantified using Kapa Biosystems Library Quantification Kit for Illumina on a Roche light cycler 96 qPCR machine. Libraries were sequenced on the Illumina HiSeq using a 250 bp paired end protocol. Reads were trimmed using Trimmomatic 0.30 with a sliding window cutoff of Q15 [61]. *De novo* assembly was performed on samples using SPAdes version 3.7 [62]. For contigs annotation, Prokka 1.11 was used [63]. NCTC8325 whole genome sequence was used as a reference for comparison. This pipeline for data analysis was optimised and performed by MicrobesNG.

### Detection of PBP2A by Western blot

*S. aureus* cells were grown to an OD$_{600}$ ~1 in 100 ml BHI. Cells were collected by centrifugation for 10 min at 5,000 rcf in a pre-chilled centrifuge. The cell pellet was resuspended in PBS and spun as before. This step was performed two more times. The pellet was resuspended in 500 μl PBS and added to pre-chilled lysing matrix tubes containing 0.1 mm acid-washed glass beads (Sigma). Cells were broken using an MP Biomedicals FastPrep 24 Homogeniser (12x, speed 6.5, 30 s). Samples were incubated on ice between each cycle. FastPrep beads were separated by a brief spin for 30 s at 13,000 rpm. The supernatant was stored at -20˚C until needed. Serum containing rabbit Anti-PBP2A antibody was incubated at room temperature with *E. coli* whole cell lysate and then with *S. aureus* SH1000 whole cell lysate to remove cross-reactive antibodies. The protein concentration of the lysates was measured using Bradford Assay (Bio-Rad) and samples were normalised to equal amounts of protein, 10 μg of total protein was separated using 12% w/v SDS-PAGE. The primary antibody (1:5000 in 1x TBST plus 1% w/v skim

milk) dilution of a rabbit anti-PBP2A polyclonal antibody was applied to membranes over-night. A horseradish peroxidase (HRP) conjugated anti-rabbit IgG secondary antibody (Sigma) (1:10,000 in 1xTBST plus 1% w/v skim milk) was incubated with the membrane for 1 hour. A chemiluminescence kit, Clarity ECL Western blotting substrates (Bio-Rad) was used for visualisation of proteins. The membrane was scanned using ChemiDoc MP Systems (Bio-Rad) for chemiluminescent detection.

## Emergence of rifampicin resistance

Bacteria were grown aerobically at 37˚C in Bacto tryptic soy broth (soybean-casein digest medium) (BD) supplemented with 5 μg/ml of oxacillin (Sigma) for MRSA strains until they reached the late logarithmic phase of growth ($OD_{625}$ ~1). The Rifampicin-resistant ($RMP^R$) isolates were selected by plating $10^9$–$10^{11}$ cells per each plate supplemented with 5 or 100 μg/ml of Rifampicin (RMP) (Sigma, cat. no. R3501) in addition to 5 μg/ml oxacillin for MRSA strains. The final number of viable cells was determined by plating an appropriate dilution on solid tryptic soy medium in the absence of RMP. Plates were incubated at 37˚C for 48 h and the number of colonies on plates with different concentrations of RMP were counted. The mutation frequencies were then calculated relative to the total number of viable cells plated.

## RNA polymerase purification

Cell lysate was diluted with TGED (10 mM Tris-HCl pH 7.9, 5% v/v glycerol, 0.1 mM EDTA, and 0.2 mM DTT) with 0.2 M NaCl to about 10 ml of buffer per 1 g of cells. Polyethylenimine was added to 0.6% v/v and centrifuged at 8000 rpm for 10 min. The pellet was then washed with TGED with 0.45 M NaCl. To elute RNAP, TGED with 1 M NaCl was added to the pellet before centrifugation at 8000 rpm for 10 min and the supernatant collected. Ammonium sulphate was then added at 0.35 g/ml of supernatant. This was then centrifuged for 30 min at 15,000 rpm. The pellet was dissolved in TGED with 0.1 M NaCl and loaded on to a HiTrap Heparin HP (GE Healthcare Life Sciences) equilibrated with TGED with no NaCl. A step elution was carried out using TGED with increasing concentrations of NaCl. Fractions containing RNAP were loaded on to a RESOURCE Q ion exchange column (GE Healthcare Life Sciences) pre-equilibrated with TGED with no NaCl. RNAP core was eluted in NaCl gradient (0.15–1 M NaCl) in TGED. Fractions were pooled, concentrated, and supplied with 50% v/v glycerol for storage. σ subunits and GreA factor were cloned from SH1000, expressed with 6xHis tags, and purified by metal-affinity chromatography using standard methods.

## *In vitro* transcription elongation assay

Oligonucleotides were from IDT. RNA was radiolabelled and complexes were assembled exactly as described [42,64] using 0.3 μM RNA, 0.7 μM template DNA, and 7 μM non-template DNA with 1 pmol RNAP per reaction in 10 μl of transcription buffer (20 mM Tris-HCl, pH 7.9, 40 mM KCl). Non-template DNA was biotinylated at the 3' end, allowing for immobilisation on streptavidin beads. Reactions were started by addition of 10 mM $MgCl_2$ with or without NTPs (GE Healthcare) to concentrations shown in figure legends, or 50 μM pyrophosphate (Sigma), or 5 μM GreA. Reactions were incubated at 37˚C for times indicated in figures and stopped with an equal amount of formamide-containing buffer. Products were resolved in denaturing (8M urea) polyacrylamide gels, visualised by phosphorimaging and analysed using ImageQuant (GE Healthcare). All reactions were repeated at least three times.

### *In vitro* transcription initiation assay

One pmol of wild-type or mutant *S. aureus* RNAP core and 5 pmols of $\sigma^A$ or $\sigma^B$ (saturating amount) and 0.2 pmols of a linear template containing sequence from -200 to +100 around the transcription start site of *pbp2, mecA and clfB*, or ASP23 genes of SH1000 or a T7A1 promoter [65,66] were mixed in 8 μl of initiation buffer (20 mM Tris acetate pH 7.9, 40 mM potassium acetate, 0.5 mM DTT, 1 μg/ml Bovine Serum Albumin (BSA), 10 mM MgCl$_2$). Transcription was initiated by the addition of 2 μl of the mixture of (final concentrations): 100 μM dinucleotide primer (IDT) corresponding to-1/+1 position of the promoter and 10 μM α-[$^{32}$P] NTP (20 Ci/mmol) (Hartmann Analytic) corresponding to +2 position. Reactions were stopped after 10-min incubation at 37˚C by the addition of formamide-containing loading buffer. Trinucleotides were separated from unincorporated radioactive NTP on 33% denaturing (8M urea) polyacrylamide gels and revealed as above. Reactions were repeated at least three times. Activities on *S. aureus* promoters were normalized to the activities on T7A1 promoter.

### Total RNA extraction and RNA-seq

Extraction and purification of *S. aureus* total RNA was carried out using a Qiagen RNeasy Plus Mini kit (Cat no. 74134). Fresh bacterial culture of OD$_{600}$ ~0.05 was grown in to OD$_{600}$ ~0.5. 8 ml of Qiagen RNAprotect Bacteria Reagent (Cat no. 76506) was added to 8 ml of culture in a 50 ml tube and incubated for 5 min at room temperature. Cells were recovered by centrifugation at 4000 rpm for 10 min at 4˚C. Pellets were stored at -70˚C. 200 μl of TE buffer (30 mM Tris-HCl, 1 mM EDTA, pH 8.0) and 200 μl proteinase K solution was added to the RNAprotect Bacteria Reagent treated pellet and vortexed. Tubes were incubated for 1 hour at 37˚C in a waterbath. The cell suspension was mixed by vortexing every 10 min for at least 10 s. Pellets were resuspended in 700 μl of RLT buffer containing β-mercaptoethanol. Cells were then disrupted using MP Biomedicals FastPrep 24 Homogeniser (3x, speed 6.5, 30 s). Cells were incubated on ice for 5 min between each cycle. The cell lysates were centrifuged for 30 s at 13,000 rpm to separate lysing matrix. Supernatant was harvested in a gDNA eliminator column and spun for 30 s at 10,000 rpm. Manufacturer's protocol was followed from here onwards. RNA concentration was measured using NanoDrop. RNA samples were stored at -4˚C for a month or -70˚C for longer period.

RNA-Seq for transcriptome analysis was performed by Glasgow Polyomics, University of Glasgow. Total RNA samples of three biological replicates for each strain were subject to QC prior to library preparation followed by ribosomal depletion using TruSeq stranded total RNA kits (Illumina). The generated raw data was compared to NCTC8325 reference sequence for expression quantification and transcript annotation. Preliminary bioinformatics support for data analysis was provided by Glasgow Polyomics which included expression quantification, statistics and differential expression analysis.

### Rate of respiration

Clark-type oxygen electrode (Rank Bros Ltd) was used to measure rate of respiration. The apparatus contains an electrode operating at a polarising voltage of 0.6 V separated by a chamber with an oxygen-permeable Teflon membrane. The reduction in the oxygen concentration was measured at the cathode which creates a potential difference. The potential difference is measured by LabTrax-4 and LabScribe2 software (World Precision Instruments). The temperature of the chamber was maintained at 37˚C and stirred at a constant rate. Prior to the experiment, the chamber was calibrated from 100% oxygen saturation to 0% oxygen by the addition of sodium dithionate. 50 ml of BHI was inoculated with a single colony and incubated overnight at 37˚C with 250 rpm shaking. Next morning, overnight culture was diluted to 1:50 in 50

ml fresh BHI and incubated to produce log phase culture for 3 h at 37˚C with 250 rpm shaking. The cultures were centrifuged at 3,380 rcf for 10 min at 4˚C and supernatant discarded. The pellet was washed twice by resuspension in ice-cold 0.02 M PBS followed by centrifugation as described above. The washed cell pellet was resuspended in 1 ml ice-cold 0.02 M PBS and kept on ice until needed. Following electrode calibration, 1950 μl of 0.02 M PBS was added to the electrode chamber and the system was left idle for 15 min to stabilise oxygen concentration inside the chamber. 50 μl of sample was added to the chamber and allowed to stabilise for 5 min. The chamber was then sealed with the lid. After 5 min, 50 μl of 1 M glucose was injected into the chamber through the lid to begin respiration. The oxygen consumption was recorded and respiration rates were calculated as $nmolO_2/min$.

## Murine sepsis models

6–8 weeks old female BALB/c mice (Charles River Laboratories, UK) were housed in designated animal facilities using standard husbandry protocols. Bacteria for inoculum were grown to stationary phase in BHI and washed 3 times in sterile endotoxin free PBS and finally resuspended in PBS containing 10% (w/v) Bovine Serum Albumin and stored at -80˚C until required. Dilutions were made to the bacterial stocks prior to injection using sterile endotoxin free PBS, with doses of 1 x $10^7$ CFU per mouse ($n$ = 10). Injections (100 μl) were made intravenously in the tail vein. Viable bacteria in the inoculum were quantified by serial dilution and plating onto BHI agar, and leaving overnight at 37˚C, before directly counting the CFU to determine injected dose. Mice were monitored and euthanised at 72 hpi unless otherwise stated or according to experimental design. Mouse organs (livers and kidneys) were individually homogenised in PBS using a Precellys 24 homogeniser. CFU per organ were determined by plating 10 μl spots of serial dilutions of the homogenate onto BHI agar for bacterial number enumeration and Statistical significance determined by Mann-Whitney two-tailed test; *ns*, $p > 0.05$; * $p < 0.05$; ** $p < 0.005$.

## Ethics statement

Murine work was carried out according to UK law in the Animals (Scientific Procedures) Act 1986, under Project License P3BFD6DB9 (*Staphylococcus aureus* and other pathogens, pathogenesis to therapy).

## Supporting information

**S1 Table. List of *S. aureus* strains associated with single copy *mecA* (*lysA*::p*mecA*).** *, denotes trained strains with intermediate oxacillin resistance (TI); †, denotes trained strains with high-level oxacillin resistance (TR); ‡, denotes TI strain trained further for high-level oxacillin resistance (TIR).
(PDF)

**S2 Table. Identification of mutations and Antibiogram of MRSA and SH1000 derived oxacillin resistant strains.** *, Only DNA sequencing was performed for the *rpoB* and *rpoC* genes; †, frameshift mutation; ‡, 11-bp insertion; [a], representative strains used in this study; [b], data taken from [1]; S, Sensitive. OX, Oxacillin; FOX, Cefoxitin; PG, Penicillin G; RIF, Rifampicin.
(PDF)

**S3 Table. List of *S. aureus* strains associated with plasmid-borne *mecA* (pRB474 p*mecA*).** *, denotes trained strains with intermediate oxacillin resistance (TI); †, denotes trained strains with high-level oxacillin resistance (TR); ‡, denotes TI strain trained further for high-level

oxacillin resistance (TIR).
(PDF)

**S4 Table. Mutations identified by whole genome sequencing in SH1000 pRB474-p*mecA*
derivatives relative to NCTC8325.**
(PDF)

**S5 Table. Mutations frequencies for rifampicin resistance in *S. aureus* strains.** Frequencies
of rifampicin-resistant colonies for strains carrying *rpoBC* mutations were determined from
plates spread with $10^9$ to $10^{11}$ viable cells supplemented with 5 and 100 µg/ml rifampicin. Frequency of RMP$^R$ mutants expressed as average (±SD) number of RMP$^R$ colonies from three
independent experiments. $<5\times10^{-11}$ corresponds to no colonies detected.
(PDF)

**S6 Table. Functional classification and differential gene expression of shared gene pools of
*mecA*$^+$ and *mecA*$^+$ *rpo* strains.** $^*$, compared against WT; $^\dagger$, compared against *lysA*::p*mecA*
(SJF4996); 26 DEGs shared by all three *rpo* mutant strains are highlighted in blue, related to
Fig 6D. Clusters of orthologues groups of proteins (COGs) were retrieved from NCBI for 121
common DEGs associated with high-level resistance where, CELLULAR PROCESSES AND
SIGNALING includes, [M] Cell wall/membrane/envelope biogenesis; [O] Post-translational
modification, protein turnover, and chaperones; [T] Signal transduction mechanisms; [V]
Defence mechanisms; INFORMATION STORAGE AND PROCESSING includes [J] Translation, ribosomal structure and biogenesis; [K] Transcription; [L] Replication, recombination
and repair; METABOLISM includes [C] Energy production and conversion; [E] Amino acid
transport and metabolism; [F] Nucleotide transport and metabolism; [G] Carbohydrate transport and metabolism; [H] Coenzyme transport and metabolism; [I] Lipid transport and
metabolism; [P] Inorganic ion transport and metabolism; [Q] Secondary metabolites biosynthesis, transport, and catabolism; and POORLY CHARACTERIZED includes [R] General
function prediction only; [S] Function unknown.
(PDF)

**S7 Table. List of strains and plasmids used in this study.** $^*$, denotes trained strains with intermediate oxacillin resistance (TI); †, denotes trained strains with high-level oxacillin resistance
(TR); ‡, denotes TI strain trained further for high-level oxacillin resistance (TIR).
(PDF)

**S8 Table. List of oligonucleotides used in this study.**
(PDF)

**S1 Fig. Oxacillin resistance and levels of PBP2A in *rpoB/C* complemented strains. A)** Oxacillin susceptibility for parental and genetically complemented strains (stated as above) were
compared using the Etest method. Oxacillin MICs are listed in brackets. **B)** The amounts of
PBP2A (~76kDa) was determined using whole cell lysates of *lysA*::p*mecA rpoB*-H929Q
(SJF5003), *lysA*::p*mecA rpoC*-G740R (SJF5034) and genetically complemented *lysA*::p*mecA
rpoB+* (SJF5044) and *lysA*::p*mecA rpoC+* (SJF5045) as well as COL and COL *rpoB+* (SJF5049)
strains.
(PDF)

**S2 Fig. Nucleotide addition and pyrophosphorolysis by SH1000, *lysA*::p*mecA rpoB*-H929Q
and *lysA*::p*mecA rpoC*-G740R RNAPs. A)** RNA was radiolabelled on the 3' end to produce
an RNA of 14 nt. Nucleotide addition resulted in the production of an RNA product of 15 nt.
(I) Schematic of scaffold before and after nucleotide addition. (II) 23% w/v polyacrylamide

denaturing gel showing nucleotide addition over time. Observed rate constants (*Kobs*) are shown below the gel (numbers that follow the ± sign are standard errors). **B)** RNA of was radiolabelled on the 5' end to produce an RNA of 13 nt. Pyrophosphorolysis first resulted in the production of an RNA product of 14 nt, subsequent pyrophosphorolysis produced shorter products. (I) Schematic of scaffold before and after pyrophosphorolysis. (II) 23% w/v poly-acrylamide denaturing gel showing pyrophosphorolysis over time. Observed rate constants (*Kobs*) are shown below the gel (numbers that follow the ± sign are standard errors).
(PDF)

**S3 Fig. Transcription elongation and pausing of SH1000, *lysA*::p*mecA rpoB*-H929Q and *lysA*::p*mecA rpoC*-G740R RNAPs.** The assembled scaffold was used to examine differences in pausing by three RNAPs during elongation. The trace of each reaction at 30 seconds on the 23% w/v polyacrylamide denaturing gels can be seen.
(PDF)

**S4 Fig. Misincorporation by SH1000, *lysA*::p*mecA rpoB*-H929Q and *lysA*::p*mecA rpoC*-G740R.** RNA of was radiolabelled on the 5' end to produce an RNA of 13 nt. Misincorporation resulted in the production of an RNA product of 15 nt. **A)** Schematic of scaffold before and after misincorporation. **B)** 23% w/v polyacrylamide denaturing gel showing misincorporation over time. Observed rate constants (*Kobs*) are shown below the gel (numbers that follow the ± sign represents standard errors).
(PDF)

**S5 Fig. RNA hydrolysis of a non-cognate nucleotide by SH1000, *lysA*::p*mecA rpoB*-H929Q and *lysA*::p*mecA rpoC*-G740R.** RNA was radiolabelled on the 5' end to produce an RNA of 15 nt. Hydrolysis resulted in the production of a radiolabelled RNA product of 13 nt. **A)** Intrinsic cleavage of mis-incorporated nucleotide by RNAPs; (I) Schematic of scaffold before and after intrinsic hydrolysis. (II) 23% w/v polyacrylamide denaturing gel showing hydrolysis over time. Observed rate constants (*Kobs*) are shown below the gel (numbers that follow the ± sign are standard errors). **B)** Gre factor-associated hydrolysis of mismatched nucleotide by RNAPs. Gre-assisted hydrolysis resulted in the production of a radiolabelled RNA product of 13 nt. (I) Schematic of scaffold before and after hydrolysis. (II) 23% w/v polyacrylamide denaturing gel showing hydrolysis over time. Observed rate constants (*Kobs*) are shown below the gel (numbers that follow the ± sign are standard errors).
(PDF)

**S6 Fig. Transcription of SH1000, *lysA*::p*mecA rpoB*-H929Q and *lysA*::p*mecA rpoC*-G740R on T7A1 in the presence and absence of oxacillin at high and low concentrations.** The terminator (Term), run off (R/O) and abortive products produced after 30 min of T7A1 incubation with the RNAPs were visualised on a 20% w/v polyacrylamide denaturing gel.
(PDF)

**S7 Fig. Schematic representation of high-level MRSA selection using multicopy plasmid-borne *mecA* and subsequent strain evolution. A)** Schematic representation of antibiotic gradient plate of two layers. Bottom layer consists of plain BHI agar, top layer supplemented with 5/20 μg/ml methicillin. **B)** Resistance properties of pRB474-p*mecA* (SJF4981) and its parental MSSA, SH1000 strain. **C)** Use of a gradient plate to select for high-level oxacillin resistance. **D)** The Etest strips revealed high-level oxacillin resistance which required presence of the pRB474-p*mecA*.
(PDF)

**S8 Fig. Schematic of the GdpP operon, showing the acquired SNPs and effect of *gdpP* deletion on resistance. A)** The genomic region of the *gdpP* operon along with *rplI* and *dnaC* genes. The N-terminus of the GdpP protein contains two transmembrane helices (black boxes), a PAS domain, GGDEF, DHH and DHHA1 domains. Amino acid substitutions identified in highly resistant derivatives of pRB474-p*mecA* (SJF4981) are indicated and strain details are shown in boxes. **B)** The inactivation of *gdpP* in Δ*gdpP*::*kan*R (SJF5025) showed susceptibility to oxacillin in the absence of *mecA*. Subsequent introduction of pRB474-p*mecA* into Δ*gdpP*::*kan*R (SJF5025) resulted in Δ*gdpP*::*Kan*R pRB474-p*mecA* (SJF5026) was accompanied by high-level resistance to oxacillin. The MIC for oxacillin determined by Etest is listed in brackets.
(PDF)

**S9 Fig. Reintroduction of *mecA* into p*mecA* cured backgrounds. A)** *gdpP*-R318L (SJF4993) was transduced with single copy *mecA* using RN4220 *lysA*::p*mecA* (SJF4994) as a donor strain for the chromosomal integration of p*mecA* into the multicopy *mecA* (pRB474-p*mecA*) cured background. **B)** Vice versa, *lysA*::*kan rpoB*-H929Q (SJF5010) was introduced with multicopy plasmid-borne pRB474-p*mecA* into the single copy *mecA* cured background. Oxacillin MICs are listed in brackets for all strains.
(PDF)

## Acknowledgments

We thank James P O'Gara (National University of Ireland, Galway) for generous gift of pRB474-p*mecA* plasmid, *S. aureus* USA300 FPR3757 and invaluable discussions, Mark Enright (Manchester Metropolitan University, UK) for *S. aureus* MRSA252, Adriana E Rosato (Institute for Academic Medicine Houston Methodist, USA) for invaluable discussions and Angelika Gründling (Imperial College London, UK) for *S. aureus* SEJ1 Δ*gdpP*::*kan*. We are also very grateful to Tracy Palmer (Newcastle University, UK) for providing us with the plasmid for Δ*SAOUHSC_00271* construction.

## Author Contributions

**Conceptualization:** Viralkumar V. Panchal, Jeffrey Green, Nikolay Zenkin, Simon J. Foster.

**Data curation:** Viralkumar V. Panchal.

**Formal analysis:** Viralkumar V. Panchal, Bohdan Bilyk, Joshua A. F. Sutton, Oliver T. Carnell, David P. Hornby, Jeffrey Green, William L. Kelley, Nikolay Zenkin.

**Funding acquisition:** Nikolay Zenkin, Simon J. Foster.

**Investigation:** Viralkumar V. Panchal, Caitlin Griffiths, Hamed Mosaei, Bohdan Bilyk, Joshua A. F. Sutton, Oliver T. Carnell.

**Methodology:** Viralkumar V. Panchal, Caitlin Griffiths, Hamed Mosaei, Nikolay Zenkin.

**Project administration:** Viralkumar V. Panchal, Nikolay Zenkin, Simon J. Foster.

**Supervision:** Jamie K. Hobbs, Nikolay Zenkin, Simon J. Foster.

**Validation:** Viralkumar V. Panchal.

**Visualization:** Viralkumar V. Panchal.

**Writing – original draft:** Viralkumar V. Panchal, Hamed Mosaei, David P. Hornby, Jeffrey Green, William L. Kelley, Nikolay Zenkin, Simon J. Foster.

**Writing – review & editing:** Viralkumar V. Panchal, David P. Hornby, Jeffrey Green, Jamie K. Hobbs, William L. Kelley, Nikolay Zenkin, Simon J. Foster.

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
