## [Decision Letter · Decision Letter 0]

18 Oct 2019

Dear Simon,

Thank you very much for submitting your manuscript "Evolving MRSA: High-level β-lactam resistance in Staphylococcus aureus is associated with RNA Polymerase alterations and fine tuning of gene expression" (PPATHOGENS-D-19-01601) for review by PLOS Pathogens. Your manuscript was fully evaluated at the editorial level and by independent peer reviewers. The reviewers appreciated the attention to an important problem, but raised some substantial concerns about the manuscript as it currently stands. Please provide additional functional data and reorganize and rewrite parts of the manuscript to improve its readibility as suggested by the reviewers. These issues must be addressed before we would be willing to consider a revised version of your study. We cannot, of course, promise publication at that time.

We therefore ask you to modify the manuscript according to the review recommendations before we can consider your manuscript for acceptance. Your revisions should address the specific points made by each reviewer.

(1) A letter containing a detailed list of your responses to the review comments and a description of the changes you have made in the manuscript. Please note while forming your response, if your article is accepted, you may have the opportunity to make the peer review history publicly available. The record will include editor decision letters (with reviews) and your responses to reviewer comments. If eligible, we will contact you to opt in or out.

(2) Two versions of the manuscript: one with either highlights or tracked changes denoting where the text has been changed; the other a clean version (uploaded as the manuscript file).

Additionally, to enhance the reproducibility of your results, PLOS recommends that you deposit your laboratory protocols in protocols.io, where a protocol can be assigned its own identifier (DOI) such that it can be cited independently in the future. For instructions see http://journals.plos.org/plospathogens/s/submission-guidelines#loc-materials-and-methods

We hope to receive your revised manuscript within 60 days. If you anticipate any delay in its return, we ask that you let us know the expected resubmission date by replying to this email. Revised manuscripts received beyond 60 days may require evaluation and peer review similar to that applied to newly submitted manuscripts.

[LINK]

Sincerely,

Andreas Peschel, Ph.D.

Associate Editor

PLOS Pathogens

François Balloux

Section Editor

PLOS Pathogens

Kasturi Haldar

Editor-in-Chief

PLOS Pathogens

orcid.org/0000-0001-5065-158X

Grant McFadden

Editor-in-Chief

PLOS Pathogens

orcid.org/0000-0002-2556-3526

Reviewer's Responses to Questions

**Part I - Summary**

Reviewer #1: This study has advanced previous work seeking to understand the mechanisms underpinning the transition from low level heterogenous to high level homogeneous methicillin resistance in MRSA. The study has been very carefully conducted and well executed. The data are very nicely presented. The focus has been on the role of RNA polymerase mutations in overcoming perturbations in anaerobic respiration associated with PBP2a expression. The authors show data on the effect of these mutations on RNA polymerase activity, specifically transcription initiation and elongation.

The role of rpo mutations in expression of homogeneous methicillin resistance has been reported before, as have experiments to investigate other mutations associated with high level resistance (all of which have been cited). This study sheds new light on the effect of rpo mutations on global transcription and respiration. This is an interesting and informative study. However, much of the data is descriptive and although a number of genes are implicated in the high level resistance phenotype associated with the rpo mutations, their precise contribution to the resistance phenotype remains unknown.

Reviewer #2: The manuscript of Panchal, et al, reports experiments to elucidate the genetic basis for expression of a homogeneous methicillin resistance phenotype from mutants selected by passage in a beta-lactam antibiotic. The investigators addressed an important limitation of prior efforts by controlling for strain background, using a mecA naïve strain SH1000, and introducing mecA under its native promoter into the chromosome as a single copy, and for comparison, on a multi-copy plasmid. The principal finding is the role of rpoB/C mutations in conferring the transition to high level resistance and in moderating the effects of introducing mecA into a naïve background. This result was shown to apply to Newman and COL, making a good case for generalizability of the observation. The authors also note the presence of an rpoC mutation, R857H, in USA300 (table S4). However, it should be noted that this strain lineage is a highly heterogeneous in its expression of methicillin resistance, so this would not seem to entirely fit with the rpoB/C mutations as facilitating homogeneous resistance; this might be worth a minor comment as to why. Unlike the results with rpo B/C, the evidence that Rex and anaerobic genes are also involved is more indirect and circumstantial, but interesting nevertheless.

In terms of general comments, the text is extremely difficult to follow at times given the large number of mutant constructs, trained and untrained, etc, and whatever can be done to reinforce reminders and remind the reader of what exactly is going on would be welcome. In addition, the experiments were conducted within the context of no regulators (mecR1-mecI or blaR1-blaI), one or both of which are almost always present in naturally occurring strains. This is a potential limitation of the generalizability of the data from these experiments.

Minor comments.

1. Lines 68-72 read: Methicillin-resistant S. aureus (MRSA) strains are not only resistant to virtually all β-lactam antibiotics but also other classes of antibiotics. This is due to the ability of MRSA to shift from being low- to high-level resistant. Prior studies revealed the involvement of chromosomal mutations for the transition of resistance but the underlying mechanism is still unknown. This is a bit misleading in that the shift from low to high-level resistance is in reference to beta-lactams not other antibiotic classes.

2. The failure of plasmid-borne mecA to confer high level resistance in the single copy pmecA cured strain (SJF5010) is interesting. Information needs to be included about the promoter for mecA on the plasmid. It is not clear why this would be the case and that the means by which the copy number and vector of mecA would select for different mechnisms is not obvious and perhaps worthy of comment.

3. Table S4. USA300 is a clone type, not a strain. Please indicate which strain. Also the abbreviations (OX, FOX, PG, RIF) lack annotation. RIF is obvious, as is OX, but the others may not be. Presumably PG = penicillin G, which if this is the case, the USA 300 strain seems to have lost its beta-lactamase plasmid and thus its regulators.

Reviewer #3: The manuscript by Panchal et al., is dedicated to characterize the molecular basis of methicillin resistance in Staphylococcus aureus. For that, two different strategies are used: a copy of the mecA gene was inserted in the genome of a methicillin-sensitive S. aureus strain or the same strain was complemented ectopically with mecA expressed from a multicopy plasmid. Then, both strains were exposed to a concentration gradient of methicillin yielding spontaneous mutants with different levels of resistance. The mutants were then sequenced and the transcript profiling analyzed. Major results/findings are: 1) high level resistance to methicillin is due to mutations in rpoB and rpoC. In some strains there are also mutations in other genes; 2) Biochemical studies indicate that mutation in RpoB and RpoC cause a deficiency in the initiation and elongation of transcription and reduce the frequency of spontaneous resistance to rifampicin; 3) methicillin highly resistant strains produce higher levels of Pbp2A, but the amount of Pbp2A does not correlate with the degree of resistance; 4) the presence of mecA is characterized by enhanced expression of genes involved in anaerobic metabolism; and 5) RpoB-H929Q restored most of the expression changes cause by the acquisition of mecA.

**Part II – Major Issues: Key Experiments Required for Acceptance**

Reviewer #1: 1. The authors have identified several genes that may be important for expression of high level resistance in strains carrying rpo mutations. To validate the study conclusions, however, it would be helpful to demonstrate experimentally that one or more of these genes is required for expression of high level resistance. For example can the high level resistance phenotype be altered by genetically manipulating expression / function of one or more of these genes?

2. The authors have done some nice experiments with COL, showing that restoration of an rpoB mutations renders this strain oxacillin susceptible (Fig. S4). Given the focus of this journal on pathogenesis and in order to further investigate the clinical significance of these data, the authors should investigate the virulence of their highly resistant strains carrying rpo mutations in comparison to the wild type strain and untrained strain carrying single mecA. Is there a change in virulence of the rpo-repaired COL strain? Many highly resistant hospital MRSA strains are relatively less virulent that MSSA or CA-MRSA strains. What are the implications of the data reported in this study for virulence.

Reviewer #2: No major issues

Reviewer #3: • The manuscript is a bit confusing and I believe that it needs to be reorganized in a simple way. For instance, the manuscript starts with the strains complemented with mecA in the muticopy plasmid. However,t the rest of the manuscript is dedicated to investigate the strains with the chromosomal copy of mecA. I believe that the originality of the manuscript comes from the approach of using strains with a single chromosomal copy of mecA. Thus I will suggest to remove the results obtained with the strains complemented with the plasmid.

• The abstract of the manuscript is very confusing and long. It needs to be rewritten completely.

• Fig 3C The authors conclude that rpoB-H929Q was impaired in initiating transcription from the SigA-dependent promoters whilst it behaved similarly at the SigB-dependent asp23 promoter. The results are not as clear because rpoB-H929Q still showed a significant inhibition in transcription initiation of asp23 promoter.

• Line 463. Regulation of gene expression in methicillin-resistant strains is largely accounted by downregulation of Rex regulator. Thus, complementation with Rex should result in the repression of Rex-regulated genes. Does complementation with Rex reduce the resistance to methicillin of trained RpoB-H929Q, trained pmecA cured rpoB-H929Q, and trained rpoC-G740R strains? is a Rex mutant resistant to methicillin?

• RpoBC mutants specifically affect the expression of certain genes (11 genes including mecA). According to the results, it is suggested that transcription pausing plays a major role in this regulation. It would be important to analyze using reporter genes fused to some of the promoters that the there are no differences in the transcription initiation.

• An important unanswered question is why strains producing rpoC-G740-R that has low transcription initiation have increased expression of mecA in vivo. It would be important to measure the mRNA levels by northern or RT-PCR and the levels of the transcription initiation by fusing the promoter of mecA to a reporter gene.

**Part III – Minor Issues: Editorial and Data Presentation Modifications**

Reviewer #1: The authors should make the manuscript results section more concise and focused. The data with the plasmid-borne mecA is not very novel (gdpP mutations associated with high level resistance have been reported before) and could be streamlined or moved to supplementary allowing the manuscript to focus more on the integrated mecA strains carrying rpo mutations. It would also be helpful to find a more accessible naming system for the strains. At present the reader has to work quite hard to follow the complicated strain names. Similarly the results section describing the transcriptional profiling is quite long and streamlines or parts moved to supplementary data.

Reviewer #2: The paper is extremely difficult to read and follow, at least for me, and supposedly I actually know the field. Whatever the authors and editors can do to simplify and clarify the message would be welcome.

Reviewer #3: • Line 210. I cannot find mutations in SAOUHSC_00591 in Table S4.

• Lines 258. Revised the text “…only applicable to the SH1000 strain a clinical MSSA isolate S. aureus Newman [37] was…”

• Line 323. Figure 3A is describe latter than Fig. 3C. Panels of Fig. 3 should be easily reorganized to follow the description of the results in the text.

• In general, it would be helpful for the reader to maintain the name of the strains all along the manuscript, for example, “Trained lysA::pmecA rpoB-H929Q (SJF5003)” is name as “Trained rpoB-H929Q” in the last section of the manuscript.

PLOS authors have the option to publish the peer review history of their article (what does this mean?). If published, this will include your full peer review and any attached files.

Reviewer #1: No

Reviewer #2: No

Reviewer #3: No

---

## [Decision Letter · Decision Letter 1]

8 May 2020

Dear Simon,

Thank you very much for submitting your manuscript "Evolving MRSA: High-level β-lactam resistance in Staphylococcus aureus is associated with RNA Polymerase alterations and fine tuning of gene expression" for consideration at PLOS Pathogens. As with all papers reviewed by the journal, your manuscript was reviewed by members of the editorial board and by several independent reviewers. The reviewers appreciated the attention to an important topic. Based on the reviews, we are likely to accept this manuscript for publication, providing that you modify the manuscript according to the review recommendations.

Sincerely,

Andreas Peschel, Ph.D.

Associate Editor

PLOS Pathogens

François Balloux

Section Editor

PLOS Pathogens

Kasturi Haldar

Editor-in-Chief

PLOS Pathogens

orcid.org/0000-0001-5065-158X

Michael Malim

Editor-in-Chief

PLOS Pathogens

orcid.org/0000-0002-7699-2064

Reviewer Comments (if any, and for reference):

Reviewer's Responses to Questions

**Part I - Summary**

Reviewer #1: The authors have improved the clarity and readability of the revised manuscript and added new data. However, while it is acknowledged that the manuscript contains a substantial amount of data, the study remains largely descriptive and does not substantially advance previous studies indicating that rpo mutations are associated with high level beta-lactam resistance in MRSA.

Reviewer #2: PPATHOGENS-D-19-01601 reports results of experiments designed to identify the genetic basis for expression of high levels of resistance (i.e., higher MICs) to oxacillin by mutants selected by passage of a methicillin-susceptible SH1000 parent strain (termed “training”) into which mecA has been introduced on a multi-copy plasmid or as a single copy with its native promoter into the chromosome. Introduction of mecA in “untrained” recipients only slightly increased toe MIC compared to the susceptible parent and the transformants phenotypically were susceptible. “Training” of these transformant by selection on methicillin greatly increased their MICs. Interestingly, the mutations associated with training of the SH1000 transformants were different depending on whether mecA was on a multi-copy plasmid or integrated as a single copy into the chromosome. When these trained mutants were cured of mecA and then mecA was re-introduced on the same vector as originally used, MICs were fully restored. On the other hand, if mecA was re-introduced in the cured, trained mutants with the other vector, resistance was not restored or only partially restored. The authors conclude that genetic backgrounds matter, but the contribution of how mecA is delivered also seems to be important and should not be dismissed. This phenomenon is not investigated further, as the investigators appropriately turn their attention to the more biologically relevant transformants containing a single copy of mecA with its native promoter, however, it might be worth a quick mention in the discussion. The results clearly demonstrate a key role of rpo mutations in trained mutants containing a single mecA gene. This is confirmed in another strain, Newman, and in a panel of clinical MRSa strains, past and extant. A plausible explanation of how these rpo mutations might confer high level resistances is provided by results of other experiments, although the fine details need to be worked out. One factor not investigated is the role of beta-lactamase and mecR1-mecI regulatory genes which are present in naturally occurring strains but not in strains used in these experiments. This might be commented upon. Overall, these experiments provide new insights into the phenotypic resistance expressed by MRSA strains and the molecular and biochemical basis for it.

1. It is not entirely clear from the results whether introduction of plasmid mecA or chromosomal mecA into SH1000 was not accompanied by any mutations besides those present on the plasmid vectors prior to “training.” This seems to be the case, but worth explicitly stating. Unless the parent and untrained mecA+ strains were not sequenced, it is not possible to conclude that the key mutations in rpo genes and gdpP were the result of antibiotic selection.

2. Is the promoter for mecA in pRB474-pmecA native or otherwise? And if not native, what is the promoter?

3. The description of results described in lines 144 through 194 is a bit hard to follow. It might be worthwhile to create a figure of a flow diagram showing the genetic manipulations, the mutants, and the MICs.

4. Please provide methods for MIC determinations. Presumably there were all Etest and no supplemental NaCl in the media.

Reviewer #3: (No Response)

**Part II – Major Issues: Key Experiments Required for Acceptance**

Reviewer #1: The authors refer to a member of the S. aureus Type VII secretion system that is required for high level resistance. This is the most novel aspect of the study and needs to be developed in order to provide new insights on the acquisition of high level beta-lactam resistance in MRSA.

Reviewer #2: No major issues

Reviewer #3: (No Response)

**Part III – Minor Issues: Editorial and Data Presentation Modifications**

Reviewer #1: (No Response)

Reviewer #2: Few...articulated in comments to authors

Reviewer #3: (No Response)

PLOS authors have the option to publish the peer review history of their article (what does this mean?). If published, this will include your full peer review and any attached files.

Reviewer #1: No

Reviewer #2: No

Reviewer #3: No
---

## [Editor Report · Decision Letter 2]

2 Jun 2020

Dear Simon,

We are pleased to inform you that your manuscript 'Evolving MRSA: High-level β-lactam resistance in Staphylococcus aureus is associated with RNA Polymerase alterations and fine tuning of gene expression' has been provisionally accepted for publication in PLOS Pathogens.

Best regards,

Andreas Peschel, Ph.D.

Associate Editor

PLOS Pathogens

François Balloux

Section Editor

PLOS Pathogens

Kasturi Haldar

Editor-in-Chief

PLOS Pathogens

orcid.org/0000-0001-5065-158X

Michael Malim

Editor-in-Chief

PLOS Pathogens

orcid.org/0000-0002-7699-2064
---

## [Editor Report · Acceptance letter]

8 Jul 2020

Dear Prof. Foster,

We are delighted to inform you that your manuscript, "Evolving MRSA: High-level β-lactam resistance in Staphylococcus aureus is associated with RNA Polymerase alterations and fine tuning of gene expression," has been formally accepted for publication in PLOS Pathogens.

Best regards,

Kasturi Haldar

Editor-in-Chief

PLOS Pathogens

orcid.org/0000-0001-5065-158X

Michael Malim

Editor-in-Chief

PLOS Pathogens

orcid.org/0000-0002-7699-2064